# A polyomavirus peptide binds to the capsid VP1 pore and has potent antiviral activity against BK and JC polyomaviruses

Joshua R Kane[1,2], Susan Fong[1], Jacob Shaul[3], Alexandra Frommlet[2], Andreas O Frank[2], Mark Knapp[2], Dirksen E Bussiere[2], Peter Kim[1], Elizabeth Ornelas[2], Carlos Cuellar[2], Anastasia Hyrina[3], Johanna R Abend[1], Charles A Wartchow[2]*

[1]Infectious Diseases, Novartis Institutes for BioMedical Research, Emeryville, United States; [2]Global Discovery Chemistry, Novartis Institutes for BioMedical Research, Emeryville, United States; [3]Chemical Biology and Therapeutics, Novartis Institutes for BioMedical Research, Emeryville, United States

**Abstract** In pursuit of therapeutics for human polyomaviruses, we identified a peptide derived from the BK polyomavirus (BKV) minor structural proteins VP2/3 that is a potent inhibitor of BKV infection with no observable cellular toxicity. The thirteen-residue peptide binds to major structural protein VP1 with single-digit nanomolar affinity. Alanine-scanning of the peptide identified three key residues, substitution of each of which results in ~1000 fold loss of binding affinity with a concomitant reduction in antiviral activity. Structural studies demonstrate specific binding of the peptide to the pore of pentameric VP1. Cell-based assays demonstrate nanomolar inhibition ($EC_{50}$) of BKV infection and suggest that the peptide acts early in the viral entry pathway. Homologous peptide exhibits similar binding to JC polyomavirus VP1 and inhibits infection with similar potency to BKV in a model cell line. Lastly, these studies validate targeting the VP1 pore as a novel strategy for the development of anti-polyomavirus agents.

*For correspondence:
charles.wartchow@novartis.com

## Introduction

BK polyomavirus (BKV), also known as human polyomavirus 1, is a small non-enveloped virus with a circular double-stranded DNA genome. BKV was first isolated from an immunosuppressed kidney transplant recipient in 1971 (*Gardner et al., 1971*), and is among the few clinically important human polyomaviruses, including JC polyomavirus (JCV) (*Padgett et al., 1971*) and Merkel cell polyomavirus (*Feng et al., 2008*). BKV is ubiquitous in human populations, with an estimated ~80% sero-prevalence worldwide (*Kean et al., 2009*; *Knowles, 2006*). Primary exposure to BKV occurs in early childhood, with 50% of 3-year-olds and over 90% of 10-year-olds testing sero-positive (*Knowles, 2001*). Post-exposure, BKV infection is characterized by subclinical persistence with kidney tissue suspected as the viral reservoir (*Ahsan and Shah, 2006*; *Heritage et al., 1981*; *Shinohara et al., 1993*). Reactivation of BKV infection can occur in conditions of immunosuppression, particularly in the context of kidney and hematopoietic cell transplantation. BKV infection in kidney transplant recipients (KTRs) is first evident as viruria (20–70% of KTRs), which can progress to viremia (10–60%); BKV nephropathy (BKVN) is diagnosed in 3–4% of KTRs and 15–50% of those patients will suffer graft loss (*Ambalathingal et al., 2017*; *Kuypers, 2012*). The primary course of care for treating BKVN is reduction of immunosuppressive therapy which carries the risk of acute graft rejection; up to 30% of BKVN cases treated by reduction of immunosuppressive therapy will experience an acute rejection episode (*Bohl and Brennan, 2007*; *Sood et al., 2012*). BKV reactivation in allogeneic hematopoietic cell transplant recipients can result in hemorrhagic cystitis (HC). In a

recent study, 16.6% of allogeneic hematopoietic cell transplantations developed HC with BKV detected in the urine in 90% of cases (*Lunde et al., 2015*). There are currently no FDA-approved antiviral therapies for BKV, presenting an unmet medical need for these indications.

The lifecycle of BKV begins with virion binding to host GT1b and GD1b ganglioside receptors (*Low et al., 2006*). The virus subsequently undergoes endocytosis via a caveolin-dependent pathway (*Eash et al., 2004*) and is trafficked in endosomes to the endoplasmic reticulum (ER) (*Jiang et al., 2009*; *Moriyama and Sorokin, 2008*), where a series of host cell enzymes orchestrate capsid disassembly (*Goodwin et al., 2011*; *Schelhaas et al., 2007*). The partially disassembled particle then interfaces with components of the ER-associated degradation (ERAD) pathway to undergo a critical step of retrotranslocation from the ER lumen into the host cell cytosol (*Bennett et al., 2013*; *Jiang et al., 2009*). Nuclear localization signal (NLS) domains within the capsid minor structural proteins then interact with components of the host nuclear pore complex to facilitate nuclear import of the viral genome (*Bennett et al., 2015*), wherein host cell transcription machinery initiates viral gene expression (*Helle et al., 2017*). The BKV virion is not known to contain any enzymes, viral or host (*Fang et al., 2010*); entry pathway steps are carried out via interactions between viral and host cell factors, and intra-virion interactions between the major and minor capsid proteins (*Zhao and Imperiale, 2017*).

The polyomavirus virion consists of a capsid formed by the major structural protein VP1 encapsidating the minor structural proteins VP2 and VP3, and the viral genome chromatinized with host histones (*Cubitt, 2006*). The capsid consists of 72 copies of homomeric VP1 pentamers cross-linked by intermolecular disulfide bonds to form a T = 7d icosahedron structure (*Nilsson et al., 2005*). VP1 pentamers within the capsid interact via *C*-terminal arms. 12 pentamers are pentavalent forming contacts with five neighbor pentamers, and the remaining 60 are hexavalent forming contacts with six neighboring pentamers (*Belnap et al., 1996*; *Prasad and Schmid, 2012*). VP1 pentamers form a central pore with five-fold symmetry, at the base of which a single copy of VP2 or VP3 is bound, forming a 5+1 complex as elucidated by X-ray and cryo-EM structures of infectious virions (*Griffith et al., 1992*; *Hurdiss et al., 2018*; *Hurdiss et al., 2016*; *Liddington et al., 1991*). All three structural proteins (VP1, VP2, and VP3) contain DNA binding domains (*Clever et al., 1993*; *Soussi, 1986*) and make contacts with the viral genome inside the infectious virion (*Carbone et al., 2004*; *Hurdiss et al., 2016*). VP2 and VP3 share a reading frame, with BKV VP3 consisting of the 232 carboxy-terminal residues of VP2 (*Helle et al., 2017*). Reconstitution of the VP1 pentamer with full-length VP2 or VP3 has yet to be achieved; reconstitution of VP1 with a truncated VP2 protein and the corresponding X-ray structure has been reported for murine polyomavirus, implicating a 'looping' structure for VP2 with the *C*-terminus interacting near the 'base', or inner virion-facing side, of the five-fold VP1 pentamer pore (*Chen et al., 1998*). Details of residues in the structural proteins contributing to the interactions of VP1 and VP2/3 have been elucidated primarily using either genetic (*Bennett et al., 2015*) or co-precipitation assays (*Barouch and Harrison, 1994*). Through these experiments, a region shared by both VP2 and VP3 near the carboxy-terminus of both proteins has been identified as required for the interaction with VP1 (*Nakanishi et al., 2006*). The biological function of the five-fold symmetry pore above the site of VP1-VP2/3 interactions at the base of capsid pentamers is not well defined. A study investigating the biological role of the JCV five-fold symmetry VP1 pentamer pore found substitution of pore residues did not interrupt VP1-VP2 interactions nor inhibit proper JCV pseudovirus (PSV) assembly; however, substitution of pore residues resulted in greatly reduced JCV PSV transduction and defective exposure of VP2 despite successful trafficking to the ER (*Nelson et al., 2015*). Due to the inhibitory nature of the five-fold VP1 pentamer pore substitutions, the authors of the aforementioned study noted the pore may be a suitable target for small-molecule antiviral therapies against polyomaviruses. For simplicity when referring to regions of the five-fold symmetry pore, 'top' indicates the region nearest the exterior of the virus, and 'bottom' or 'lower' indicate the region nearest the interior of the virus.

In the current study, we report the discovery of a thirteen amino acid BKV VP2/3-derived peptide D1$_{min}$ (corresponding to VP2 residues 290–302) that binds to BKV VP1 pentamers with single-digit nanomolar $K_D$. We show that homologous peptide derived from JCV VP2/3 binds JCV VP1 with similar affinity, demonstrating a conserved binding interface. Protein-observed 2D NMR studies show this peptide interacts with VP1 residues in a previously uncharacterized location within the five-fold symmetry pore formed by pentameric VP1, with the binding location within the pore further corroborated by a structurally-guided model generated using X-ray data from co-complexed D1$_{min}$ and

VP1. Treatment of cells with $D1_{min}$ in the context of BKV infection elicits nanomolar antiviral activity by the peptide, with a relationship established between peptide binding affinity to VP1 pentamers and antiviral potency by alanine-substitution of key peptide residues. Additionally, we show the peptide exhibits antiviral activity against JCV, potentially indicating a pan-antiviral mechanism. We demonstrate through cell-based assays that the antiviral mechanism of action (MoA) involves blocking key steps in the viral entry pathway, likely prior to the critical step of ER-to-cytosol retrotranslocation. Mutations of residues in the VP1 pore that mediate peptide binding or of residues in the VP2/3 region from which the peptide is derived impact BKV infectivity, indicating the peptide-binding site may constitute a previously uncharacterized VP1-VP2/3 binding interface. In short, we report the first anti-BKV and anti-JCV molecule that directly targets the polyomavirus VP1 five-fold symmetry pentamer pore.

## Results

### VP2/3-derived peptide binds pentameric capsid protein VP1 with high affinity

In order to better characterize the structural relationship between the BKV major structural protein VP1 and the minor capsid proteins VP2 and VP3, we focused on a stretch of amino acids near the carboxyl-terminus of VP2/3 previously referenced as 'D1' (*Nakanishi et al., 2006*). The region is largely invariant in related polyomaviruses JCV and simian virus 40 (SV40) (*Figure 1A*; *Figure 1—figure supplement 1*). We initially tested binding of a 22-mer peptide $VP2_{281-302}$ (APGGANQRTA PQWMLPLLLGLY; $D1_{22}$) to purified BKV VP1 pentamers ($VP1_{2-362}$) by Biacore surface plasmon resonance (SPR) and measured high affinity binding to the pentamer ($K_D$ = 4.8 nM; *Figure 1B,C*; *Table 1*). The curve from a 1:1 interaction model overlays well with the SPR data (*Figure 1C*), consistent with a high quality, specific interaction despite the hydrophobic nature of this peptide. In addition to directly measuring binding affinity by SPR, we developed an AlphaScreen assay to detect binding of carboxy-terminus biotinylated $D1_{22}$ to VP1 and measure the half-maximal concentration at which the biotinylated peptide is displaced by unlabeled peptide ($IC_{50}$) (*Figure 1D*, *Table 1*). The $IC_{50}$ for unlabeled $D1_{22}$ is 11 ± 2.9 nM and this value is comparable to the SPR-determined $K_D$. We additionally tested the homologous $D1_{22}$ sequence from JCV with its cognate VP1 pentamer, and observed an $IC_{50}$ of 44 ± 6.4 nM (*Figure 1D*, *Supplementary file 1*). Noting that protein-protein interactions often involve 'molecular hot spots' where most of the binding energy is associated with a limited number of interactions (*Van Roey et al., 2014*), we split $D1_{22}$ into two fragments, $VP2_{281-290}$ (APG-GANQRTA) and $VP2_{290-302}$ (APQWMLPLLLGLY, henceforth referred to as $D1_{min}$), and tested each fragment for binding to VP1. While no binding was observed for $VP2_{281-290}$ up to 10 µM (data not shown), we observed similar binding affinity for the 13-mer peptide $D1_{min}$ as was observed for $D1_{22}$ ($VP2_{290-302}$; $K_D$ = 1.4 ± 0.49 nM, $IC_{50}$ = 3.6 ± 0.57 nM) (*Figure 1D*, *Table 1*). Hereafter, references to amino acid positions in $D1_{min}$ will be based on their sequence position in VP2.

### Alanine-substitutions in $D1_{min}$ peptide reveal key residues contributing to $D1_{min}$-VP1 interaction

To identify key residues involved in the interactions of $D1_{min}$ with BKV VP1, we performed alanine-scanning mutagenesis (*Cunningham and Wells, 1989*) on $D1_{min}$, substituting one residue per peptide (*Figure 1B*), and analyzed the effect on binding to VP1 pentamers by Biacore SPR and the AlphaScreen competition assay (*Figure 1E*; *Table 1*). The SPR $K_D$ and biochemical assay $IC_{50}$ results are comparable for the alanine-substituted peptides (*Table 1*). Both assays identify peptide residues W293, L297, and L298 as key determinants of high affinity binding, with each substitution causing ~600–1000 fold loss of affinity to VP1 ($K_D$ = 920 ± 190 nM, 1600 ± 610 nM, and 1000 ± 160 nM, respectively). Alanine-substitution of other residues (M294, L295, L299, Y302) results in 40–60 fold loss of affinity ($K_D$), demonstrating that these may also contribute to binding affinity.

To see if we could further reduce the size of the peptide required for high affinity binding, we evaluated hexamer peptides covering the length of $D1_{min}$ in the AlphaScreen displacement assay (*Figure 1F*; *Table 1*). All hexamer peptides were significantly less potent relative to $D1_{min}$. Notably, peptide $D1_{min}$ HEX4 ($_{293}$WMLPLL$_{298}$) contains all three key determinant residues and yet has an $IC_{50}$ that is greater than 1000-fold higher than that of full-length $D1_{min}$ peptide. Trimer peptides covering

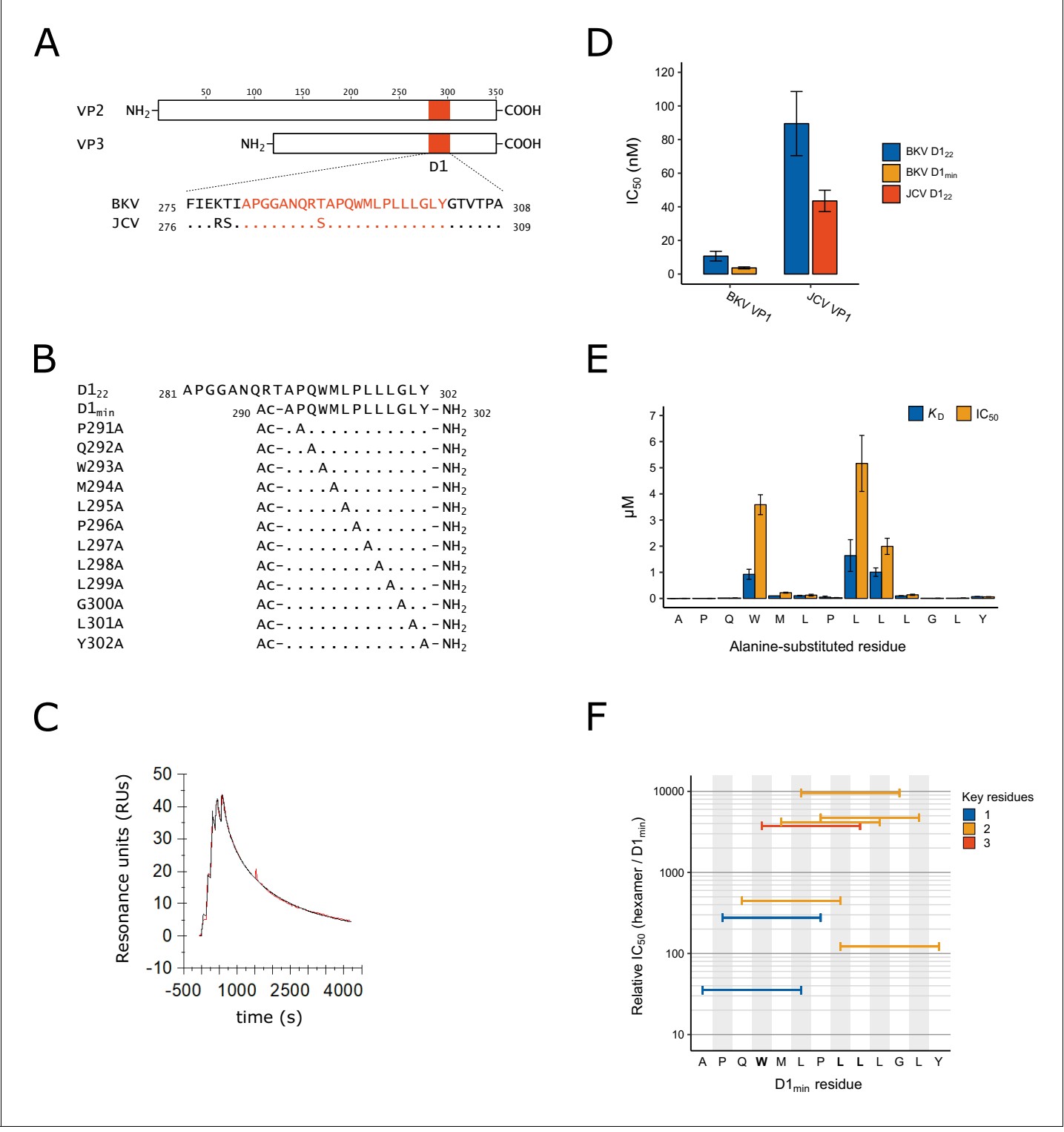

**Figure 1.** Identification of $D1_{min}$ peptide and key residues contributing to interaction with VP1. (**A**) Multiple sequence alignment of VP2/3 D1 region and flanking sequence. BKV: BK polyomavirus; JCV: JC polyomavirus. (**B**) Sequence and index within BKV VP2 of peptides used in this study, highlighting alanine-scanning mutagenesis. Ac: acetyl group (**C**) Representative surface plasmon resonance (SPR) sensorgram of single-cycle kinetic experiment showing association of $D1_{min}$ with VP1 pentamer. Multiple (five) injections are shown, and dissociation of the peptide starts at peak response. Experimental data (red) and the 1:1 model of responses (black) are shown. (**D**) Results of AlphaScreen competitive binding assay. Displacement of carboxy-terminal biotinylated $D1_{22}$ peptide from either BKV or JCV VP1 was assayed using $D1_{22}$ (BKV and JCV) or $D1_{min}$ (BKV only), with $IC_{50}$ concentration determined (mean ± SD, n = 3 for BKV VP1, n = 2 for JCV VP1). (**E**) SPR-measured VP1 binding affinity ($K_D$) and AlphaScreen

*Figure 1 continued on next page*

*Figure 1 continued*

displacement assay results (IC$_{50}$; mean ± SD, n = 3) for single-site alanine-substitutions in D1$_{min}$. (F) AlphaScreen displacement assay IC$_{50}$ values for D1$_{min}$ hexamer peptides (mean of n = 3). Color indicates the number of key residues (W293, L297, or L298) present in the hexamer.

The online version of this article includes the following figure supplement(s) for figure 1:

**Figure supplement 1.** Multiple sequence alignments of polyomavirus VP1 and VP2.

the length of D1$_{min}$ yielded similar results, showing greater than 1000-fold reductions in binding affinity to VP1 relative to the full-length D1$_{min}$ peptide (*Supplementary file 2*). We conclude that the key D1$_{min}$ residues W293, L297, and L298 contribute significantly to the interaction of D1$_{min}$ and VP1; however, additional peptide residues are required for the highest affinity binding.

## Peptide D1$_{min}$ binds within the upper pore of VP1 pentamers

In order to determine the binding location of the D1$_{min}$ peptide to VP1 pentamers, protein-observed 2D-NMR spectroscopy was performed. $^{1}$H,$^{13}$C-HMQC spectra of $^{2}$H,$^{12}$C-BKV VP1$_{30-297}$ with $^{1}$H,$^{13}$C methyl-labeled residues Ile- (I), Leu- (L), Val- (V) and Thr- (T) were recorded in the absence and presence of increasing amounts of wild-type or W293A D1$_{min}$ peptides, and ligand-induced chemical shift perturbations (CSPs) and line broadening were monitored. To enable mapping of binding locations we obtained peak assignments for select methyl groups through a combination of amino acid point mutations and a 4D-NOESY-HSQC based methyl walk (*Proudfoot et al., 2016*).

**Table 1.** Peptide IC$_{50}$ and $K_D$ measurements.

Values are mean ± SD where applicable (AlphaScreen: n = 3; SPR: n = 2). ND: Not determined.

| Peptide Name | Sequence | AlphaScreen VP1 IC$_{50}$ (nM) | SPR VP1 $K_D$ (nM) | Relative IC$_{50}$ (vs D1$_{min}$) | Relative $K_D$ (vs D1$_{min}$) |
|---|---|---|---|---|---|
| D1$_{22}$ | APGGANQRTAPQWMLPLLLGLY | 11 ± 2.9 | 4.8 | 2.8 | 3.6 |
| D1$_{min}$ | Ac-APQWMLPLLLGLY-NH$_2$ | 3.6 ± 0.57 | 1.4 ± 0.49 | 1.0 | 1.0 |
| D1$_{min}$ P291A | Ac-A**A**QWMLPLLLGLY-NH$_2$ | 3.6 ± 0.16 | 3.6 | 1.0 | 2.9 |
| D1$_{min}$ Q292A | Ac-AP**A**WMLPLLLGLY-NH$_2$ | 15 ± 1.2 | 18 | 3.8 | 13 |
| D1$_{min}$ W293A | Ac-APQ**A**MLPLLLGLY-NH$_2$ | 3600 ± 380 | 920 ± 190 | 900 | 660 |
| D1$_{min}$ M294A | Ac-APQW**A**LPLLLGLY-NH$_2$ | 220 ± 18 | 100 | 55 | 71 |
| D1$_{min}$ L295A | Ac-APQWM**A**PLLLGLY-NH$_2$ | 130 ± 34 | 110 ± 15 | 32 | 75 |
| D1$_{min}$ P296A | Ac-APQWML**A**LLLGLY-NH$_2$ | 27 ± 7.9 | 51 ± 39 | 6.8 | 36 |
| D1$_{min}$ L297A | Ac-APQWMLP**A**LLGLY-NH$_2$ | 5200 ± 1100 | 1600 ± 610 | 1300 | 1200 |
| D1$_{min}$ L298A | Ac-APQWMLPL**A**LGLY-NH$_2$ | 2000 ± 310 | 1000 ± 160 | 500 | 720 |
| D1$_{min}$ L299A | Ac-APQWMLPLL**A**GLY-NH$_2$ | 140 ± 24 | 97 ± 11 | 35 | 69 |
| D1$_{min}$ G300A | Ac-APQWMLPLLL**A**LY-NH$_2$ | 8.5 ± 0.83 | 9.3 | 2.0 | 6.4 |
| D1$_{min}$ L301A | Ac-APQWMLPLLLG**A**Y-NH$_2$ | 14 ± 1.6 | 11 | 3.5 | 7.9 |
| D1$_{min}$ Y302A | Ac-APQWMLPLLLGL**A**-NH$_2$ | 62 ± 6.5 | 70 ± 3.8 | 16 | 50 |
| D1$_{min}$ HEX1 | Ac-APQWML-NH$_2$ | 140 ± 44 | ND | 35 | ND |
| D1$_{min}$ HEX2 | Ac-PQWMLP-NH$_2$ | 1100 ± 270 | ND | 280 | ND |
| D1$_{min}$ HEX3 | Ac-QWMLPL-NH$_2$ | 1800 ± 400 | ND | 450 | ND |
| D1$_{min}$ HEX4 | Ac-WMLPLL-NH$_2$ | 15000 ± 4500 | ND | 3800 | ND |
| D1$_{min}$ HEX5 | Ac-MLPLLL-NH$_2$ | 17000 ± 3500 | ND | 4200 | ND |
| D1$_{min}$ HEX6 | Ac-LPLLLG-NH$_2$ | 38000, >40000, >40000 | ND | 9600 | ND |
| D1$_{min}$ HEX7 | Ac-PLLLGL-NH$_2$ | 19000 ± 2500 | ND | 4700 | ND |
| D1$_{min}$ HEX8 | Ac-LLLGLY-NH$_2$ | 490 ± 120 | ND | 120 | ND |

The NMR peaks assigned to BKV VP1 residues T224, T226, V231 (pro-*R* and pro-*S*), and V234 (pro-*R* and pro-*S*) were greatly affected by the addition of the peptides (*Figure 2A*). These residues are in close proximity to each other, clustering in the upper pore of the VP1 pentamer (*Figure 2B*). Binding of peptides to the upper pore appears specific since the site can be saturated (no additional CSPs observed at higher peptide concentrations) and only certain VP1 residues show CSPs while the majority of signals remained unaffected. As the set of perturbed peaks and the directions of chemical shifts are the same for both the wild-type and the alanine-substituted peptide, it can safely be concluded that both ligands have the same binding pose. Interestingly, while the wild-type peptide induces strong line broadening of certain peaks at sub-stoichiometric ligand concentrations, an observation that can be attributed to slow exchange kinetics, the alanine-substituted peptide causes pure chemical shift changes, which are usually a sign of fast chemical exchange (*Figure 2A*, top right corner). These results are consistent with our SPR and biochemical assay data, which showed that the wild-type $D1_{min}$ peptide has a high affinity interaction with VP1 (low nanomolar $K_D$) whereas the W293A peptide interacts with weaker affinity ($K_D$ >1 μM).

To validate the interaction of peptide $D1_{min}$ within the VP1 pore, we focused on three sets of residues proximal to the observed CSPs, P232, V234, and T224/T243, and tested $D1_{min}$ binding to VP1 proteins with substitutions at these residues using our AlphaScreen assay (*Figure 2C*). We found that non-polar to polar substitution of either P232 or V234 leads to a substantial decrease in peptide binding relative to wild-type VP1 (P232S: 1.3 ± 0.26%, V234S: 1.6 ± 0.32% of wild-type signal). In contrast, a substitution that conserved the hydrophobicity of the putative binding site increased observed peptide binding (V234I: 250 ± 26% of wild-type signal). Alanine-substitution of VP1 pore residues further down into the pore (T224A/T243A) impact binding to a lesser degree (34 ± 0.56% of wild-type signal). X-ray structures of VP1 proteins with these substitutions do not appear to have any major structural rearrangements (*Figure 2—figure supplement 1A,B*) and we observe normal pentamer formation of these VP1 variants by size-exclusion chromatography during purification (data not shown). Consistent with our observations, previous work on JCV five-fold symmetry VP1 pentamer pore residues showed substitution of multiple pore residues, including the homologous residue for BKV VP1 P232, did not affect either pentamer formation or thermal stability, with X-ray data showing structural differences from wild-type VP1 pentamers were largely limited to the pore itself (*Nelson et al., 2015*). These results are consistent with a specific interaction of peptide $D1_{min}$ within the upper pore of VP1 pentamers.

## NMR identifies second peptide binding site at the base of VP1

At a wild-type $D1_{min}$ peptide concentration of 25 μM, well above the low nanomolar $K_D$ observed for the primary peptide binding site in the upper pore, we observed additional ligand induced CSPs and line broadening in the 2D NMR spectrum. The VP1 peaks that were affected upon addition of peptide and for which assignments were available are I45, T46, T118, T238, and T243 (*Figure 2—figure supplement 1A–B*). Without exception, these residues are located in the lower pore of the VP1 pentamer. Based on the first appearance of spectral changes at 25 μM $D1_{min}$ peptide for a titration starting at 6.25 μM (where no CSPs were observed), we estimate that the $K_D$ is greater than 250 μM. For peptide $D1_{min}$ W293A, only at a ligand concentration of 100 μM did a few very weak additional peak shifts became visible; hence, the $K_D$ value for this peptide is likely in the low single-digit millimolar range. At a wild-type peptide concentration of 50 μM and higher the above-mentioned peaks as well as signals from amino acids located in the upper pore show significant line broadening (*Figure 3—figure supplement 1C*). This observation can potentially be explained by binding of multiple ligand copies or small soluble peptide aggregates. However, as the peptide induces signal perturbations of only certain VP1 residues all of which are in close proximity, the interaction site likely represents a binding hotspot. Consistent with this result, the predicted location of the second interaction site is consistent with the modeled position of the D1 region of BKV VP2/3 in a recent cryo-EM structure (*Figure 2E Hurdiss et al., 2018*).

## X-ray model details interaction between $D1_{min}$ peptide and VP1 pore

To further characterize the binding mode of $D1_{min}$ within the VP1 pore, a structurally guided model was generated using 2.36 Å resolution X-ray data from 13-mer $D1_{min}$ peptide in complex with truncated $VP1_{30-297}$ pentamers (*Supplementary file 3*). The VP1 pentamer model is in good agreement

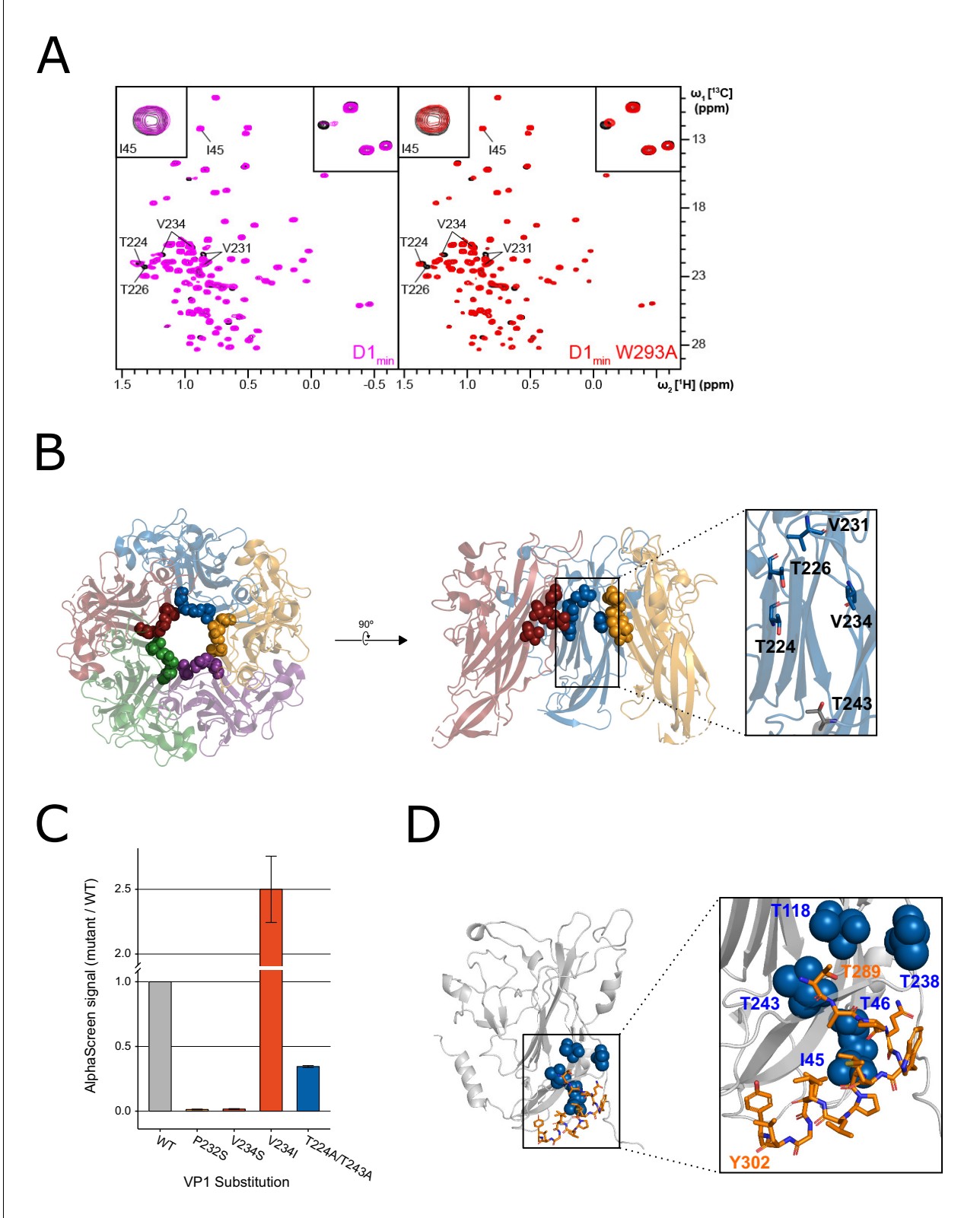

**Figure 2.** NMR characterization of the VP1-D1$_{min}$ interaction. (**A**) $^1$H,$^{13}$C-HMQC spectra showing peptide (12.5 μM) induced perturbations of tr-VP1 (125 μM; black) ILVT methyl signals. Left: the wild-type D1$_{min}$ peptide (magenta) causes CSPs and line broadening of peaks clustered in the upper pore of the target protein. The disappearance of peaks indicates slow exchange kinetics and thus, strong (usually sub-micromolar) binding (see inset in upper right corner). At sub-stoichiometric peptide concentrations no binding to a second site is observed as there are no changes of I45 (see inset in

*Figure 2 continued on next page*

*Figure 2 continued*

upper corner). Right: alanine-substituted W293A peptide (red) induces the same CSP pattern as the wild-type peptide, however, exchange kinetics are fast and no line broadening is observed. There is also no second site binding observed at low peptide concentrations. (B) VP1 residues highlighted in (A) overlaid on X-ray structure of VP1 pentamer, looking down into the pore (left) and a cutaway side-view of three VP1 monomers (right). Spheres highlight VP1 residues that exhibit CSPs upon peptide binding (T224, T226, V231, and V234). Residue T243 is lower in the pore (shown in gray) and does not exhibit perturbations upon peptide binding (PDB: 4MJ1; *Neu et al., 2013*). (C) Relative binding affinities of $D1_{22}$ peptide to wild-type VP1 protein or VP1 containing pore residue substitutions using AlphaScreen detection method. Values are normalized to wild-type VP1 (mean ± SD). (D) Overlay of 'second-site' VP1 residues (I45, T46, T118, T238, T243; blue) on cryo-EM model of BKV VP1 (gray) and VP2 (orange) (adapted from PDB 6ESB, *Hurdiss et al., 2018*).

The online version of this article includes the following figure supplement(s) for figure 2:

**Figure supplement 1.** Chemical shift pertubations induced by $D1_{min}$ binding to VP1.
**Figure supplement 2.** Structural comparison of wild-type BKV VP1 with pore residue substitution variants.

with a previously published BKV VP1 pentamer structure (PDB: 4MJ1; *Neu et al., 2013*) (RMSD: 0.85 Å; *Figure 3—figure supplement 1D*). Electron density for the peptide is observed in the upper third of the VP1 pentamer pore, consistent with the NMR binding data (*Figure 3A*, *Figure 3—figure supplement 1C*). Refinement with a best-fit model of observed electron density maps yields a primary chain of density consistent with an α-helical peptide running *N*-terminus at the top of the pentamer pore to *C*-terminus lower in the pore (*Figure 3B*), although electron density maps indicate multiple binding poses of the helix within the pore. VP1 pore residues that show peptide-induced CSPs by 2D NMR (T226, V231, V234; *Figure 2A–B*) as well as residues important for peptide binding as determined by substitution (P232, V234; *Figure 2C*) form a hydrophobic pocket around key $D1_{min}$ residues L297 and L298 (*Figure 3C*). Interestingly, pocket structure appears to be largely unaltered by ligand binding (*Figure 3—figure supplement 1D*). In conclusion, our structurally-guided model of $D1_{min}$ in complex with VP1 agrees with NMR, alanine-scan, and pore residue substitution studies placing the peptide in the upper pentamer pore and highlighting the importance of $D1_{min}$ residues L297 and L298, as well as VP1 residues T226, V231, P232, and P234 in peptide binding.

## $D1_{min}$ peptide is a potent anti-BKV inhibitor

After observing high affinity binding of the $D1_{min}$ peptide to the VP1 pentamer pore, we asked whether the peptide could inhibit BKV infection in a cell-based assay. Primary renal proximal tubule epithelial (RPTE) cells were pre-treated with a titration of peptide for 2 hr then challenged with infectious BKV (isolate MM), with indirect immunofluorescent staining for large T-Antigen (TAg) measured 48 hours post-infection (h.p.i) as a readout for productive delivery of the viral genome to the nucleus. We observed potent antiviral activity from $D1_{min}$ with a half-maximal effective concentration ($EC_{50}$) of 30 ± 6.6 nM without observable cytotoxicity in the concentration range tested (*Figure 4A, B*). Importantly, single alanine-substituted peptides of $D1_{min}$ showed a loss of antiviral activity concordant with their loss of VP1 affinity in in vitro binding assays (W293A:>5000 nM, L297A:>5000 nM, Y302A: 280 ± 51 nM). These results appear to validate the pore as an antiviral target.

As the VP1 pore region and VP2 D1 region demonstrate high sequence identity between BKV and the related human polyomavirus JCV (*Figure 1—figure supplement 1*), we tested $D1_{min}$ antiviral properties on JCV. COS-7 cells were subjected to synchronized infection by either BKV or JCV (isolate Mad-1) followed by treatment with a titration of $D1_{min}$ peptide, with indirect immunofluorescent staining for VP1 measured 72 h.p.i as a readout for productive delivery of the viral genome to the nucleus (*Figure 4C*). We observe similar $EC_{50}$ values for both polyomaviruses (BKV: 220 nM, JCV: 350 nM), albeit roughly 10-fold higher than observed for BKV in the RPTE cell model. This may be due to potential differences in entry pathway between human RPTE and simian COS-7 cells as is observed with related CV-1 cells (*Bennett et al., 2013*), or higher viral titer required to infection COS-7 cells relative to RPTE cells.

We next asked if $D1_{min}$ treatment showed MOI-dependence, noting that if the virus is the peptide target the observed $EC_{50}$ should shift to a higher concentration due to the increase in target abundance. We observe a significant shift (15.4-fold) in $EC_{50}$ from low to high MOI (*Figure 4D*), consistent with the model that the virus is the target for $D1_{min}$ antiviral activity and more inhibitor is required with increasing viral challenge.

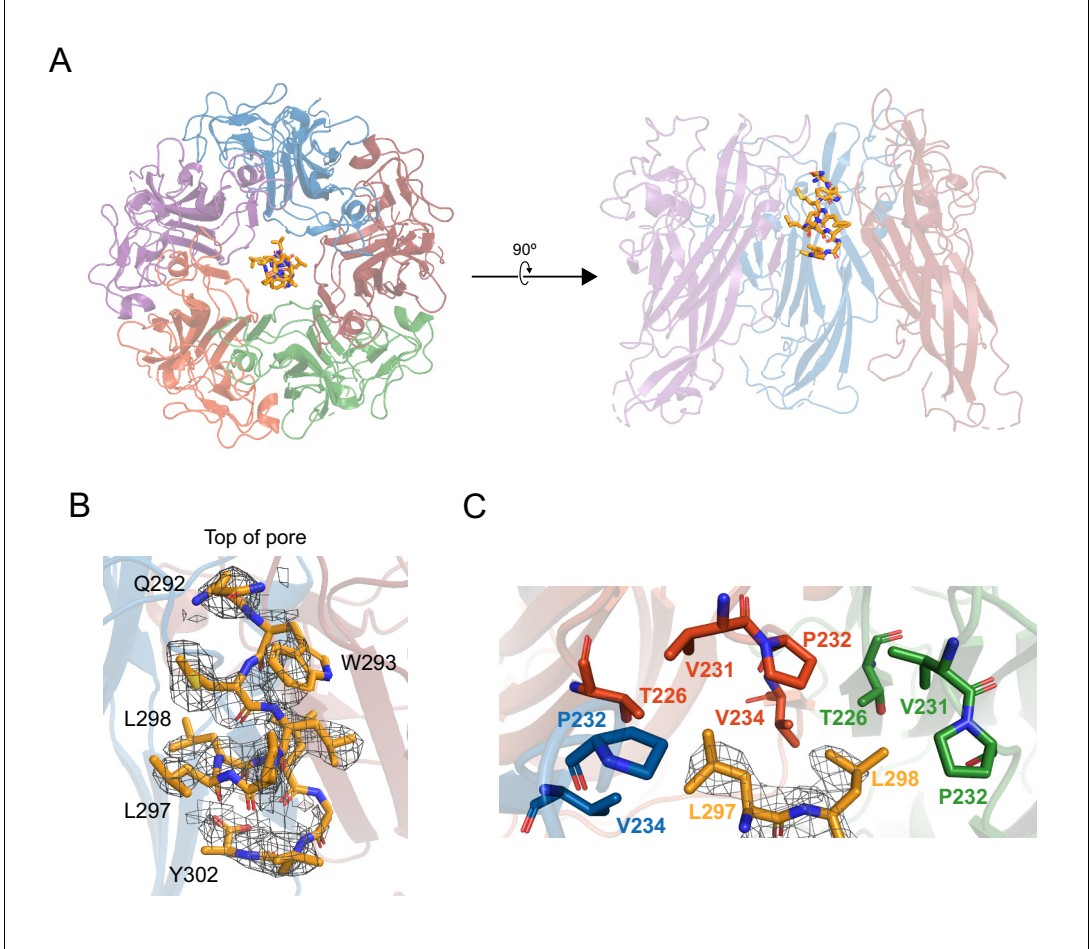

**Figure 3.** X-ray structurally-guided model of D1$_{min}$-VP1 pentamer complex shows key residues mediating interaction. (A) Structurally-guided model of structure of D1$_{min}$ peptide bound to BKV VP1 pentamer. (Left) Top-down view of the model. (Right) Cutaway representation showing three VP1 molecules of the pentamer with D1$_{min}$ peptide bound. (B) D1$_{min}$ 2Fo-Fc electron density map, contoured at 1σ, with model of peptide residues $_{292}$QWLPLLLGLY$_{302}$ built with guidance from the experimental maps. Start and end residues, as well as key binding residues W293, L297, and L298 are highlighted. (C) Close-up of hydrophobic pocket formed by VP1 pore residues T226, V231, P232, and V234. Blue, orange, and green residues represent three distinct VP1 molecules within the pentamer. D1$_{min}$ electron density for residues L297 and L298 (yellow), shown contoured to 1σ, correspond to regions of closest approach of the peptide to the pocket.

The online version of this article includes the following figure supplement(s) for figure 3:

**Figure supplement 1.** Additional description of X-ray structurally-guided model of D1$_{min}$ peptide in complex with VP1 pentamer.

A notable difference between infectious BKV virions and VP1 pentamers or virus-like particles (VLPs) containing only VP1 is the presence of minor structural proteins at the base of the VP1 pentamer pore (*Hurdiss et al., 2016*). Based on the structural studies presented in *Figures 2* and *3*, the proposed mechanism of antiviral action by D1$_{min}$ is through binding of the peptide to the VP1 pore. To confirm that D1$_{min}$ peptide can bind to infectious BKV virions containing the minor structural proteins VP2 and VP3, we performed an affinity purification of biotinylated D1$_{22}$ in the presence of VP1 moieties. Full-length VP1 pentamers, VLPs, or purified infectious BKV virions were incubated in 10-fold molar excess of either biotinylated or unlabeled D1$_{22}$, followed by affinity-purification of biotinylated peptide and assaying co-purification of VP1. VP1 pentamers, VLPs, and infectious particles co-purified with D1$_{22}$, demonstrating that the peptide can bind to infectious BKV virions (*Figure 4E*). Interestingly, only amino-terminal biotinylation was compatible with the assay; carboxy-terminal D1$_{22}$ was unable to co-purify VP1 pentamers or VLPs, even when tested with truncated VP1 pentamers (VP1$_{30-297}$) and extended peptides (*Figure 4—figure supplement 1*, *Supplementary file 2*). These

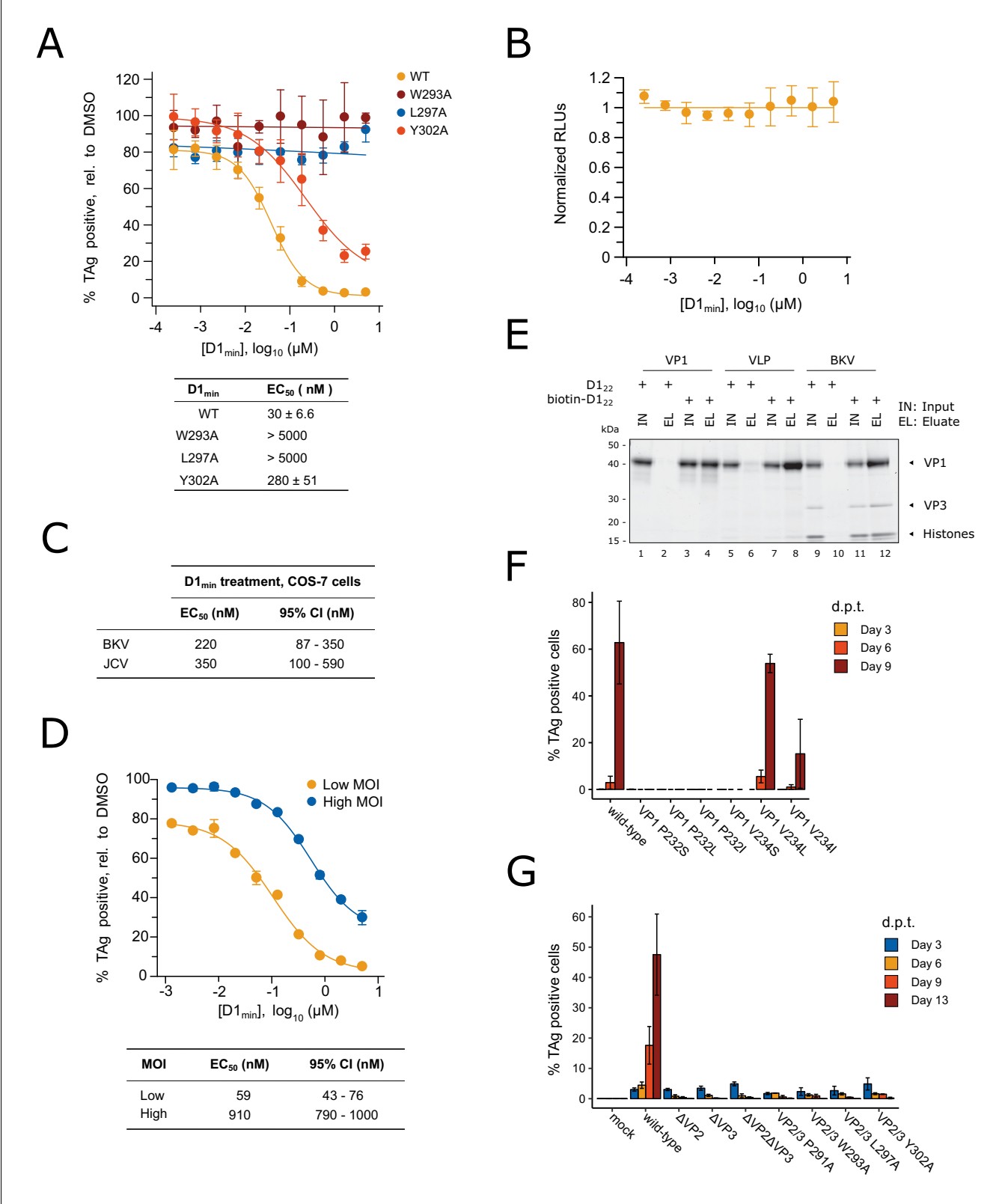

**Figure 4.** D1min peptide has nanomolar antiviral activity. (A) Dose-response curves for wild-type D1min peptide and three alanine-substituted variants (W293A, L297A, L298A) in single-round BKV infection assay in RPTE cells (mean ± SD, n = 3), and table of derived EC50 values. Productive delivery of the viral genome to the nucleus is quantified by fraction of RPTE cells expressing BKV TAg by indirect immunofluorescent staining 48 hr post-infection (h.p. i). (B) CellTiter-Glo luminescent cell viability assay to measure D1min cytotoxicity in RPTE cells after two days of treatment. Relative light units (RLUs) are

*Figure 4 continued on next page*

*Figure 4 continued*

normalized to DMSO treatment (mean ± SD, n = 2). (C) $D1_{min}$ $EC_{50}$ values with 95% confidence intervals (CI) are shown for single-round infection assay of JCV and BKV in COS-7 cells, measuring fraction of VP1 expressing cells 72 h.p.i. (D) Dose-response curves for wild-type $D1_{min}$ peptide in low (0.3) versus high (5) BKV MOI, measuring fraction of TAg-positive cells 48 h.p.i. $EC_{50}$ values with 95% confidence intervals (CI) are shown. (E) Coomassie-stained gel showing streptavidin purification of VP1 pentamers, BK VLPs, or infectious BKV virions using either $D1_{22}$ or biotinylated-$D1_{22}$ peptide. (F) BKV spreading infection assay with VP1 pore mutants, measuring TAg-positive cells 3, 6, and 9 days post-transfection of BKV genomic DNA. d.p.t.: days post-transfection (mean ± SD, n = 2). (G) Same as (F), with BKV VP2/3 mutants (mean ± SD, n = 3). While residue position is relative to VP2, VP2/3 indicates mutation is present in both proteins.

The online version of this article includes the following figure supplement(s) for figure 4:

**Figure supplement 1.** $D1_{22}$ peptide requires *N*-terminal biotinylation to co-purify BKV VP1.
**Figure supplement 2.** Expression of mutant BKV VP1 and VP2/3 proteins.

data are consistent with the X-ray structurally-guided model placing the *N*-terminus of the peptide at the top of the VP1 pore. We conclude $D1_{min}$ peptide can bind to infectious BKV virions that contain minor structural proteins at the base of VP1 pores and importantly, that the in vitro biophysical and biochemical characterization of $D1_{min}$ peptide-VP1 pore binding can provide a model for understanding peptide-BKV interactions and $D1_{min}$-associated antiviral activity.

## VP1 pore single-point mutants result in loss of BKV infectivity

As the D1 region of VP2/3 contains the same amino acid sequence as $D1_{min}$, we tested whether residues that mediate $D1_{min}$ binding to the VP1 pore are important for BKV infectivity. We performed site-directed mutagenesis of BKV *VP1* in the context of the viral genome, introducing substitutions at two key peptide binding residues in the VP1 pore, P232 and V234, and performed a spreading infection assay. Circularized wild-type or mutant BKV genomes were transfected into RPTE cells and productive, spreading infection was monitored by indirect immunofluorescent staining of expressed TAg over a time course of 3, 6, and 9 days post-transfection (d.p.t.) (*Figure 4F*). We observe robust spreading infection for wild-type BKV by 9 d.p.t. In contrast, BKV was completely intolerant of all tested substitutions at P232, as was previously observed in the homologous residue P223 in JCV (*Nelson et al., 2015*), as well as substitution V234S. V234L did not appear to affect BKV infectivity, and V234I, which showed increased binding to biotinylated peptide in an AlphaScreen biochemical assay, exhibited an intermediate phenotype with incomplete inhibition of viral spread. Importantly, all mutant viruses expressed similar levels of VP1 to wild-type BKV (*Figure 4—figure supplement 2A*), dismissing interpretations that the observed phenotypes are due to differences in VP1 expression. While we cannot determine at what stage of the viral lifecycle the pore mutations are affecting viral infectivity (e.g. during assembly versus during entry), previous work with JCV pore mutants demonstrated no effect on JCV PSV assembly or VP2 association with VP1 (*Nelson et al., 2015*). Next, we performed site-directed mutagenesis on BKV *VP2/3* in the context of the viral genome and repeated the spreading infection assay (*Figure 4G*). While wild-type and mutant BKV all expressed TAg at similar levels 3 d.p.t. after transfection, only wild-type BKV exhibited a spreading infection in culture. BKV was completely intolerant of VP2 or VP3 deletion, and of all tested alanine-substitutions within the D1 region of VP2/3 — no detectable infectious virus produced from these mutant genomes. This is despite observing no significant impact on VP2/3 expression levels in mutants VP2 W293A and VP2 L297A (*Figure 4—figure supplement 2B*). We conclude that residues involved in the VP1-$D1_{min}$ interaction observed in vitro are required for productive BKV infection.

## $D1_{min}$ peptide requires interaction with BKV for activity, but does not block viral endocytosis

Past studies have utilized broadly acting inhibitors of cellular activities to interrogate the polyomavirus entry pathway (*Goodwin et al., 2011*; *Moriyama and Sorokin, 2008*; *Ravindran et al., 2017*; *Schelhaas et al., 2007*). Such studies have been coupled with time-of-addition assays, in which treatment with inhibitors is initiated at different times during infection to correlate an inhibitor mechanism of action with a particular stage of BKV entry, including endocytosis (*Eash et al., 2004*), endosome maturation and vesicular trafficking (*Eash and Atwood, 2005*; *Jiang et al., 2009*), and ERAD/proteasome activity (*Bennett et al., 2013*). Similarly, we conducted a time-of-addition assay to better characterize at which stage of the BKV entry pathway $D1_{min}$ antiviral activity occurs. RPTE

cells were subjected to a synchronized infection at low multiplicity of infection (MOI) and inhibitor was added at varying times post-infection, with productive delivery of the BKV genome to the nucleus assessed by indirect immunofluorescent staining of TAg expression at 48 h.p.i. (*Figure 5A*). In addition to treatment with D1$_{min}$, we treated infected cells with an anti-BKV neutralizing monoclonal antibody P8D11 (*Abend et al., 2017*) and cell-penetrating TAT-fused modifications (*Vivès et al., 1997*) of D1$_{min}$ which exhibit similar antiviral activity and biochemical potency to untagged D1$_{min}$ peptide (*Supplementary file 2* and *Supplementary file 4*). We observe a nearly complete loss of D1$_{min}$ antiviral activity by 4 h.p.i. (*Figure 5B*), consistent with the timing of viral endocytosis (*Eash et al., 2004*). The BKV neutralizing antibody P8D11 parallels the time-dependent loss of activity of D1$_{min}$. Cell-penetrating variants of D1$_{min}$ show delayed loss of activity compared to the unmodified peptide, with only an approximate 50% loss of activity at 4 h.p.i. and a gradual tapering off of activity in subsequent timepoints. For comparison, previous time-of-addition work using Brefeldin A and nocodazole, treatments which affect viral trafficking to the ER, showed efficacy against BKV until 10–12 h.p.i (*Jiang et al., 2009*).

Our extensive biochemical and biophysical characterization of the D1$_{min}$ peptide showing specific, high-affinity binding to the five-fold symmetry VP1 pentamer pore and the observed relationship between VP1 binding affinity potency and antiviral efficacy in alanine-substitution peptides implicates an antiviral MoA involving direct binding to BK virions. However, an interesting alternative model exists wherein D1$_{min}$ directly interacts with a putative host-cell receptor to induce an antiviral signaling pathway resulting in the observed antiviral activity associated with peptide treatment. To exclude this possibility, we first tested whether or not treatment of RPTE cells with D1$_{min}$ peptide induced expression of known antiviral genes, comparing induction to treatment with interferon beta (IFN-β) (*Figure 5—figure supplement 1*). While we observe robust expression of all genes tested with 20 hr of IFN-β treatment (except for IFNA2, which showed modest increase in expression after IFN-β treatment), we fail to observe gene induction with D1$_{min}$ peptide treatment. Next, we tested whether peptide can bind to the surface of RPTE cells, a requirement for interacting with a host-cell receptor. We assessed peptide-cell binding using *N*-terminal biotinylated D1$_{22}$ peptide, assessing peptide binding to the cell surface by immunofluorescent microscopy and using the addition of BK VLPs as a positive control for recruiting peptide to the cell surface (*Figure 5—figure supplement 1*). We observe background biotin staining when cells were treated only with biotinylated D1$_{22}$ peptide and observe a large increase in biotin staining intensity with the addition of BK VLP, with the two stains overlapping to a high degree. Importantly, the increase in biotin stain is abrogated upon addition of unlabeled D1$_{min}$ WT peptide in excess, demonstrating the recruitment of peptide to the cell surface by BK VLPs is specific; the same abrogation is not observed when using unlabeled D1$_{min}$ L297A peptide. We conclude that there is no evidence to support a model of D1$_{min}$ directly acting on host cells, and that our data is consistent with a peptide antiviral mechanism requiring direct interaction with BKV virions.

After observing loss of D1$_{min}$ activity rapidly after initiating BKV infection in our time-of-addition study, we next asked whether D1$_{min}$ acts as a cell-binding antagonist by performing a cell binding assay. Briefly, RPTE cells were incubated with BKV that had been pre-treated with D1$_{min}$ or P8D11 for 1 hr at 4°C to block endocytosis. Cells were rinsed, immediately fixed, and we performed indirect immunofluorescent staining for cell-associated VP1 puncta as a readout for cell-bound virions (*Figure 5C*). We observed no effect on the number of cell-associated VP1 puncta in cells treated with D1$_{min}$ up to 5 μM (>100 fold over observed EC$_{50}$). In contrast, we observe loss of cell-associated VP1 puncta in the presence of the neutralizing antibody P8D11 treatment starting at concentrations >0.43 nM (>62 ng/mL), roughly the observed EC$_{50}$ concentration. In addition, we tested whether pre-treatment of the virus, as performed in this assay, has any impact on antiviral efficacy of D1$_{min}$ peptide and observed no difference in EC$_{50}$ between pre-treatment of BKV versus pre-treatment of RPTE cells with D1$_{min}$ peptide (*Figure 5—figure supplement 3*). We conclude that D1$_{min}$ does not block binding of BKV to cells and the observations from the time-of-addition experiment are due to the inability of the peptide to permeate the cell membrane rather than the peptide inhibiting BKV adsorption. This model is consistent with the delayed loss of antiviral activity observed for cell-penetrating variants of D1$_{min}$. Thus co-entry with the infecting virion, consistent with direct binding to the five-fold symmetry VP1 pentamer pore, is required for D1$_{min}$ entry into cells and subsequent antiviral activity.

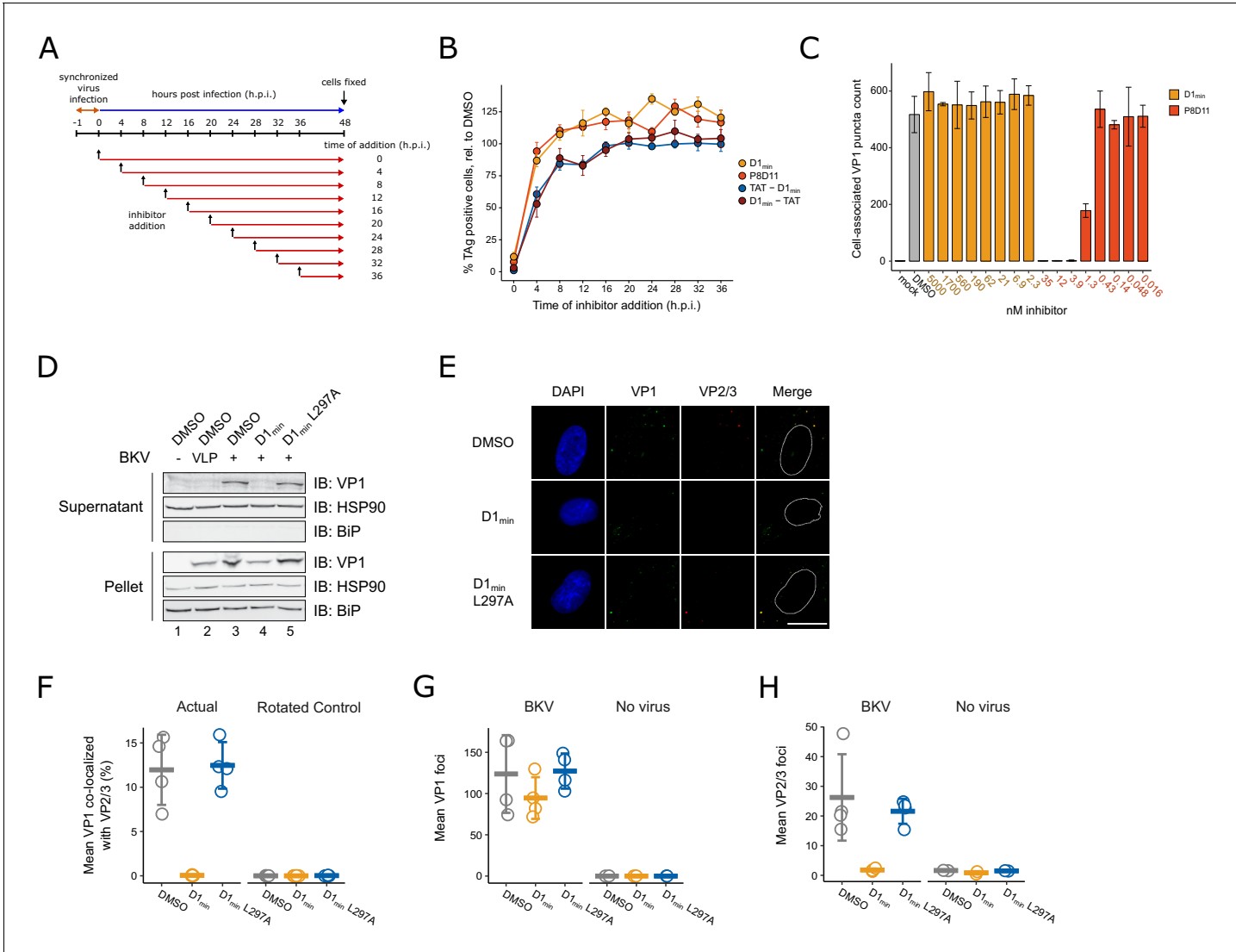

**Figure 5.** D1$_{min}$ inhibits key steps in virion processing during entry. (**A**) Schematic of time-of-addition assay. (**B**) Time-of-addition assay with D1$_{min}$, cell-penetrating peptides TAT-D1$_{min}$ and D1$_{min}$-TAT, and anti-BKV neutralizing antibody P8D11. BKV infected cells were treated with inhibitors at 10-fold over measured EC$_{50}$ concentrations. Productive delivery of the viral genome to the nucleus is measured by the fraction of RPTE cells expressing BKV TAg by indirect immunofluorescent staining 48 hr post-infection (h.p.i), relative to DMSO-treated samples (mean ± SD, n = 4). (**C**) Virus cell binding inhibitor assay. BKV was treated with indicated inhibitor at indicated concentrations for 1 hr on ice, adsorbed to cells for 1 hr at 4˚C, unbound virus washed away, and remaining cell-associated virus measured by indirect immunofluorescent staining of VP1 (mean ± SD, n = 3). (**D**) ER-to-cytosol retrotranslocation assay. RPTE cells subjected to a synchronized BKV infection (high MOI), cells were harvested 24 h.p.i, and lysates were fractionated into a supernatant (cytoplasmic) and pellet fraction. Fractions were then analyzed by SDS-PAGE, and VP1 protein and cellular compartment markers were detected by immunoblotting. (**E**) Representative microscopy images of VP2/3 exposure assay. Minor capsid proteins were detected using a polyclonal antibody able to recognize both VP2 and VP3. Scale bar: 20 µm. (**F**) Quantification of images exemplified in (**E**), measuring fraction of VP1 stain co-localizing with VP2/3 stain, averaged per well. Rotated data indicate calculated co-localization between VP1 and VP2/3 stains after rotating VP2/3 images 90˚ to assess rate of random association between the two (mean ± SD, n = 4). (**G**) Quantification of VP1 foci in images exemplified in (**E**), averaged per well (mean ± SD, n = 4 for infected samples, n = 2 for uninfected samples). (**H**) Quantification of VP2/3 foci in images exemplified in (**E**), averaged per well (mean ± SD, n = 4 for infected samples, n = 2 for uninfected samples).

The online version of this article includes the following figure supplement(s) for figure 5:

**Figure supplement 1.** RT-qPCR assay for induction of innate immunity-associated genes.

**Figure supplement 2.** D1$_{22}$ peptide does not bind to RPTE cells in the absence of BK VLPs.

**Figure supplement 3.** D1$_{min}$ antiviral potency is unaffected by pre-treatment of cells versus pre-treatment of virus.

## $D1_{min}$ activity occurs prior to BKV ER-to-cytosol retrotranslocation

Next, we examined whether $D1_{min}$ treatment affects the ER-to-cytosol retrotranslocation of BKV, a critical entry step and distinguishing feature of polyomaviruses (*Dupzyk and Tsai, 2016*). This transition can be assayed by fractionation of infected host cells and testing for the presence of VP1 protein in the cytosolic fraction (*Bennett et al., 2013*; *Inoue and Tsai, 2011*). RPTE cells were subjected to a synchronized BKV infection at high MOI followed by treatment of $D1_{min}$ peptide (wild-type and L297A) at 10-fold over $EC_{50}$ concentration. BKV VLPs were included as a negative control as they are unable to cross from the ER lumen into the cytosol (*Geiger et al., 2011*). At 24 h.p.i, cells were harvested, partially permeabilized with digitonin, fractionated between supernatant (cytosol) and insoluble pellet (e.g. ER, nucleus), and subjected to reducing SDS-PAGE followed by immunoblotting to detect the presence of VP1 in each fraction (*Figure 5D*). We only observe the presence of VP1 protein in the cytosolic fraction for untreated (DMSO) or $D1_{min}$ L297A-treated BKV samples. Samples treated with wild-type $D1_{min}$ or lacking the minor structural proteins VP2/3 (BK VLP) have no detectable VP1 protein in the cytosolic fraction, indicating the virus is unable to proceed through this step of the viral lifecycle. In contrast, we observe the presence of VP1 in the pellet fraction for all samples with BK VLP or BKV, indicating the virus (or VLP) has undergone endocytosis under all treatments. We conclude that $D1_{min}$ peptide antiviral activity against BKV occurs after endocytosis, but prior to retrotranslocation of the virus from the host cell ER to the cytosol.

## $D1_{min}$ peptide inhibits exposure of minor structural proteins during capsid disassembly

During polyomavirus entry, inter-pentamer disulfide bonds are oxidized by host enzymes resulting in dissociation of pentavalent VP1 pentamers from the capsid (*Kuksin and Norkin, 2012*; *Schelhaas et al., 2007*), exposing minor structural proteins VP2/3 to immunostaining (*Norkin et al., 2002*). Notably, inhibition of VP2/3 exposure is indicative of improper trafficking or disassembly of the virus (*Bennett et al., 2013*). We asked if $D1_{min}$ affected this step of the BKV lifecycle. RPTE cells were subjected to a synchronized BKV infection at high MOI followed by treatment of $D1_{min}$ (wild-type and L297A) at 10-fold over $EC_{50}$ concentration. At 24 h.p.i., cells were fixed and stained by indirect immunofluorescence against VP1 and VP2/3 (*Figure 5E*). To assess the ratio of infectious particles to total particles in the cells, we calculated the fraction of virion particles (VP1-stained puncta) that co-localized with VP2/3 stain (*Figure 5F*). We observe a pronounced loss of VP1 co-localized with VP2/3 in wild-type $D1_{min}$ treated cells as compared to our untreated control. In contrast, no change in co-localization between VP1 and VP2/3 is observed for treatment with the loss-of-function peptide $D1_{min}$ L297A. Consistent with this result, overall VP2/3 staining was reduced by treatment with $D1_{min}$ peptide only, whereas VP1 staining was unchanged across all treatments (*Figure 5G–H*). We conclude that treatment with $D1_{min}$ peptide results in virions that are unable to proceed through proper capsid disassembly that would result in the essential exposure of minor structural protein epitopes.

## Discussion

Polyomaviruses are the causative agents of multiple human diseases, and the lack of effective antiviral therapeutics for the treatment of polyomavirus infections and associated diseases represent an unmet medical need. Accordingly, we report the discovery and characterization of the first described anti-polyomavirus inhibitor that acts through a novel antiviral MoA by binding to the viral capsid five-fold symmetry VP1 pentamer pore – the BKV VP2/3-derived peptide $D1_{min}$. Treatment with $D1_{min}$ elicits potent antiviral activity against BKV in cell-based assays ($EC_{50}$ = 30 nM). $D1_{min}$ antiviral activity is also observed against related JCV, denoting a possible conserved MoA.

To characterize the $D1_{min}$ antiviral MoA, we performed informative, exploratory mechanistic studies. Time of inhibitor addition studies showed $D1_{min}$ must co-enter the cell with BKV to elicit antiviral activity, since $D1_{min}$ lost antiviral efficacy when administered after viral endocytosis. Importantly, time-of-addition studies with a cell-penetrating version of $D1_{min}$, a TAT-fused peptide, resulted in a delayed loss of activity relative to unmodified peptide, which is consistent with an antiviral mechanism occurring within the cell. In agreement with an intracellular MoA, $D1_{min}$ did not block binding of BKV to host cells. Fractionation experiments identified the $D1_{min}$ MoA as occurring prior to the

critical step of BKV retrotranslocation from the ER lumen to the host cell cytosol, and the presence of BKV capsid protein in the pelleted fraction is consistent with a $D1_{min}$ MoA that does not block BKV entry into cells.

Through extensive biophysical (SPR), biochemical (AlphaScreen), and structural (NMR, X-ray crystallography) characterization of $D1_{min}$, we show the peptide specifically and unambiguously binds to a novel site in the upper pore formed by VP1 pentamers with high affinity (SPR $K_D$ = 1.4 nM, biochemical $IC_{50}$ = 3.6 nM). Alanine-scanning of $D1_{min}$ identified key residues that mediate interaction of $D1_{min}$ with VP1, with ~1000 fold loss of affinity for VP1 associated with single residue substitutions, demonstrating that peptide binding is specific and not due to non-specific binding driven by peptide hydrophobicity. Moreover, key $D1_{min}$ residue substitutions that result in loss of VP1 affinity also result in loss of $D1_{min}$ antiviral efficacy, linking VP1 binding and antiviral activity. Consistent with this link between in vitro binding studies and in-cell BKV assays, we show a related variant, the biotinylated $D1_{22}$ peptide, can bind infectious virions using in vitro co-purification studies.

With these findings, we posit the following model for $D1_{min}$ antiviral activity: $D1_{min}$ binds to the upper pore of VP1 pentamers in the capsid of BKV prior to endocytosis; peptide remains bound to the virus during endocytosis, and antiviral activity occurs within the cell after endocytosis but prior to escape from the host ER lumen, resulting in inhibition of VP2/3 exposure. A caveat to this proposed MoA is the lack of direct testing of whether peptide remains engaged with BKV within the cell after endocytosis, which would further inform an intracellular mechanism directly involving association of the peptide with VP1. This mechanism, or other mechanisms involving, for example, intracellular release of the peptide and interaction with an intracellular receptor, are interesting and may be differentiated with additional study.

We examined an alternate mechanism where the peptide binds to a cell-surface receptor and induces a pathway that results in antiviral activity. To explore this possibility, we performed several experiments that differentiate this MoA from our proposed MoA. In summary, we do not observe peptide association with the cell surface in the absence of VP1 protein, we do not observe induction of a panel of antiviral genes after cell treatment with $D1_{min}$ peptide, and we observe an MOI-dependence of $D1_{min}$ antiviral efficacy. In the last experiment, higher viral titers require more peptide to bind the increased abundance of VP1 pore binding sites, whereas the abundance of a putative cell-surface receptor should be independent of viral titer. Furthermore, the observed relationship between loss of VP1 binding affinity and loss of antiviral efficacy by alanine-substitution of key $D1_{min}$ residues is consistent with the proposed MoA; that a putative cell-surface receptor would bind the same peptide sequence and also be susceptible to the same alanine-substitutions as the VP1 binding site is unlikely, but cannot be excluded. Additional evidence that is contrary to the MoA involving a cell-surface target is as follows: one would not expect the TAT-fused peptides to behave differently from unmodified peptide in the time-of-addition studies if the $D1_{min}$ target is on the cell surface. $D1_{min}$ and TAT-fused $D1_{min}$ share the same binding sequence and have similar $IC_{50}$ values in our biochemical assay (TAT-$D1_{min}$: 2.7 nM, $D1_{min}$-TAT:<1.5 nM, $D1_{min}$: 3.6 nM). As both *N*-terminal and *C*-terminal TAT-fused $D1_{min}$ showed a delayed loss of activity relative to unmodified peptide, it is unlikely that the TAT addition is disrupting an interaction with a putative cell-surface receptor. While we cannot definitively rule out alternative $D1_{min}$ MoAs that indirectly impact BKV infection, we find that the current studies are consistent with the MoA where the $D1_{min}$ peptide initially binds to the VP1 pentamer pore of BKV, it crosses the cell-membrane via endocytosis of BKV with associated $D1_{min}$, and has anti-viral activity within the cell.

We characterized the binding determinants of $D1_{min}$ as well as demonstrated binding specificity of the peptide to VP1 pentamers using alanine-scanning mutagenesis (*Cunningham and Wells, 1989*; *Lozano et al., 2017*). We identified three key residues in $D1_{min}$ – W293, L297, and L298 – where single alanine-substitutions resulted in ~1000 fold loss of binding affinity to VP1 as measured in both an AlphaScreen competition assay and SPR studies. A structurally-guided model based on X-ray data acquired from VP1-$D1_{min}$ complexes details an α-helical peptide binding in the upper VP1 pore, running *N*- to *C*-terminal from the top of the pore to the lower pore. Additionally, the model places key $D1_{min}$ residues L297 and L298 within a hydrophobic pocket formed by VP1 pore residues T226, V231, P232, and V234. We note that the homologous residue to P232 in JCV was identified as a critical residue in VP1 pore biology (*Nelson et al., 2015*). We observe that these residues experience chemical shift perturbations (CSPs) by 2D NMR upon peptide binding to VP1, with the exception of P232, which is not labeled with $^{13}C$ isotope and is therefore not visible by this NMR method.

Biochemical peptide binding assays with substitutions of VP1 pore residues P232 and V234, as well as BKV spreading infection studies with mutations at VP1 P232 or V234, confirm and validate both the 2D NMR data and structurally-guided model of the peptide binding in the VP1 pore. Studies of trimer and hexamer truncations of the 13-mer $D1_{min}$ peptide were unable to reconstitute high-affinity binding, including a hexamer peptide that contained the three key residues W293, L297, and L298, indicating these key residues are necessary but not sufficient for high affinity peptide binding to VP1. This may indicate that other residues likely contribute to peptide potency and/or the helical conformation of the peptide is important for its mode of binding. Peptides less than nine residues in length are unlikely to form secondary structures (*Gellman, 1998*; *Manning et al., 1988*). Accordingly, the high affinity binding of $D1_{min}$ to VP1 likely uses some combination of structural and sequence elements.

Since the $D1_{min}$ peptide is derived from a native sequence found in the VP2/3 D1 region, and the peptide binds the five-fold symmetry VP1 pentamer pore with high affinity, intriguing questions are raised about the biology of the peptide and the pore to which it binds. Mutations in pore residues that are in close proximity to the peptide-binding site including VP1 P232 and V234 generally result in noninfectious virus without grossly altering VP1 pentamer structure. Likewise, all tested mutations in the D1 region of BKV VP2/3 resulted in noninfectious virus. We have shown that pore residues VP1 T226, V231, P232, V234 are important for peptide binding and that the D1 region of VP2/3 from which the $D1_{min}$ peptide is derived, namely VP2/3 290–302, are largely invariant across multiple polyomaviruses (*Figure 1—figure supplement 1*). Viral proteins are subject to rapid sequence change over time due to relatively high viral genome mutagenesis rates unless maintained by purifying selective pressure (*Daugherty and Malik, 2012*; *Duffy et al., 2008*; *Kistler et al., 2007*; *Tokuriki et al., 2009*); polyomaviruses are no exception (*Buck et al., 2016*; *Pastrana et al., 2013*). As such, invariance of viral protein sequences is a potential indication of biological relevance. Further to this point, studies with JCV show both high affinity peptide binding to VP1 and inhibition of infection at a similar potency ($EC_{50}$) as BKV in COS-7 cells, suggesting a conserved mechanism across at least two polyomaviruses. Both the sequence invariance of residues in the VP1 pore and in the VP2/3 D1 region that mediate $D1_{min}$-VP1 interactions, and the convergence of phenotype between the VP1 pore mutations and VP2/3 D1 region mutations suggest that the peptide-binding site represents a previously uncharacterized VP2/3 binding site in the VP1 pore.

Multiple structures have been reported previously for infectious polyomavirus virions using both X-ray crystallography and cryo-electron microscopy (cryo-EM), with most reporting the minor structural proteins VP2/3 as a globular density at the base of the VP1 pentamer (*Griffith et al., 1992*; *Hurdiss et al., 2016*; *Liddington et al., 1991*). More recently, a cryo-EM structure of an infectious BKV virion mapped the resolved *C*-terminus of VP2/3, including the region containing the $D1_{min}$ sequence, at the base of the VP1 pentamer (*Hurdiss et al., 2018*). These results are consistent with our current observations of a $D1_{min}$ low affinity 'second binding site' identified in the NMR studies. Although the affinity of 13-mer peptide for the 'second site' is low, in the intact virion the D1 region that matches the sequence of $D1_{min}$ may bind this site with higher affinity, particularly in the context of full-length VP2/3.

An intriguing question raised by the study is whether the specificity and high affinity of $D1_{min}$ binding to VP1 is coincidental or reconstitutes an interaction between VP1 and the VP2/3 D1 region that occurs during the viral lifecycle. Dramatic restructuring of the viral capsid takes place during disassembly including reduction of intra-capsid disulfide bonds, decrease in virion size, and exposure of previously hidden minor structural protein epitopes from the capsid core (*Bennett et al., 2013*; *Geiger et al., 2011*; *Inoue and Tsai, 2011*; *Jiang et al., 2009*; *Magnuson et al., 2005*; *Norkin et al., 2002*). Our BKV infection assays indicate that residues involved in $D1_{min}$ binding to the VP1 pore are critical for BKV infectivity. The low- and high-affinity $D1_{min}$ binding sites found at the base and in the upper pore of VP1 pentamers, respectively, may represent VP1-VP2/3 interactions during different stages of disassembly, and subsequent exposure of VP2/3 may be required to allow for essential viral-host protein interactions. For example, interaction of the VP2/3 NLS with importin α/β is required for efficient nuclear import of SV40 and BKV viral genomes during entry (*Bennett et al., 2015*; *Nakanishi et al., 1996*; *Nakanishi et al., 2002*). The process by which polyomaviruses decrypt masked VP2/3 protein-protein interaction domains via structural rearrangements may involve the translocation of VP2/3 D1 region from a low-affinity binding site at the base of VP1 pentamers to the high-affinity binding site in the upper pore. Further studies are required to validate

the proposed model, in particular studies that may deconvolute potential VP2/3 interactions with the lower and upper VP1 pore.

The essential entry step of membrane penetration by non-enveloped viruses is an incompletely characterized process (*Kumar et al., 2018*). Polyomaviruses complete this lifecycle step by exploiting the host cell ER-associated degradation (ERAD) pathway to undergo retrotranslocation from the ER lumen into the cytosol (*Dupzyk and Tsai, 2016*). This process involves viral interaction with numerous host factors and requires the presence of VP2/3 (*Bagchi et al., 2016*; *Bagchi et al., 2015*; *Dupzyk et al., 2017*; *Geiger et al., 2011*; *Inoue and Tsai, 2017*). Treatment of BKV with D1$_{min}$ peptide inhibits two key observable lifecycle steps: exposure of VP2/3 epitopes to immunofluorescent staining and retrotranslocation of VP1 protein from the ER lumen into the cytosol. The mechanism by which D1$_{min}$ elicits these antiviral observations remains incompletely defined. Interestingly, mutations in the VP1 pore of JCV PSVs reduce exposure of VP2/3 (*Nelson et al., 2015*), consistent with a D1$_{min}$ antiviral mechanism directly elicited via pore binding. In addition, VLPs which lack minor structural proteins or polyomavirus with VP2 or VP3 deletions are unable to escape from the ER lumen (*Geiger et al., 2011*; *Inoue and Tsai, 2011*), and this translocation step is also blocked for BKV in the presence of D1$_{min}$. Accordingly, we envision a few, non-exclusive models for D1$_{min}$ antiviral action. First, D1$_{min}$ binding in the VP1 pore may block interactions with host factors that interact with the pore. Indeed, a cryo-EM study of infectious BKV particles identified possible heparin binding to the upper VP1 pore (*Hurdiss et al., 2018*). Blocking host factor interactions could result in improper trafficking of the incoming virion or inhibit capsid processing. Previous studies have shown that small molecules that inhibit polyomavirus trafficking to the ER block exposure of VP2/3 (*Bennett et al., 2013*; *Bennett et al., 2015*). Second, binding of D1$_{min}$ peptide may stabilize the capsid and inhibits proper disassembly. The host restriction factor HD5 was shown to both inhibit VP2/3 exposure and stabilize the JCV capsid (*Zins et al., 2014*), suggesting that the two phenotypes may be related. Third, D1$_{min}$ may disrupt an essential interaction between the VP1 pore and VP2/3. Last, D1$_{min}$ may block an interaction between a host factor and VP2/3. As D1$_{min}$ and VP2/3 D1 region share an identical sequence, the peptide may act by dissociating from the VP1 pore after endocytosis and binding to a host factor VP2/3 binding site, blocking an essential interaction. In general, these models posit an antiviral mechanism of antagonist D1$_{min}$ peptide acting as a competitive inhibitor for the VP1 pore binding site, the VP2/3 D1 region, or both in the case of disrupting a putative interaction between VP2/3 and the VP1 pore. Deciphering which, if any, of these models is correct will require further investigation.

In conclusion, to identify potential therapeutics for BK and JC polyomaviruses, we explored the potential of targeting the major capsid protein VP1, one of the few proteins expressed by members of the polyomavirus family. This strategy of antiviral agents targeting viral capsids has also been explored for HIV, dengue virus, picornaviruses, and hepatitis B virus (*Blair et al., 2010*; *Byrd et al., 2013*; *De Colibus et al., 2014*; *Deres et al., 2003*; *Fox et al., 1986*; *Klumpp et al., 2018*; *Lamorte et al., 2013*). Relevant to polyomaviruses, a study of the JCV VP1 pentamer pore confirmed the sensitivity of the virus to substitution of pore residues, establishing the potential of the pore as a target for small-molecule therapies (*Nelson et al., 2015*). D1$_{min}$ represents the first antiviral agent against BK and JC polyomaviruses that specifically targets the VP1 pentamer pore. The peptide is derived from polyomavirus minor structural proteins VP2/3 D1 region, and NMR and X-ray studies show the peptide binds to a novel site within the pore formed by pentameric VP1 capsid protein. The biological relevance of the interaction between peptide and the VP1 pore was tested by mutagenesis of the viral genome, with cell-based viral proliferation assays being impacted by mutations within the VP2/3 D1 region or the corresponding binding region within the VP1 pore. These observations indicate the peptide-binding site may be biologically-relevant, potentially constituting a previously uncharacterized VP1-VP2/3 binding interface. Given the single-digit nanomolar binding affinity of D1$_{min}$ to the VP1 pore, the peptide provides a powerful new tool molecule for probing polyomavirus entry biology. In particular, as the *N*-terminus of the peptide faces the exterior of the virion, one could imagine extending the peptide in this direction to add an assay-dependent probe to monitor stages of BKV entry, including virion disassembly and viral-host interactions. Finally, the D1$_{min}$-binding site within the VP1 pore reported in this study may represent a novel target for development of first-in-class antiviral therapies to address the unmet medical need presented by polyomavirus infections.

## Materials and methods

### Cell culture

Primary renal proximal tubule epithelial (RPTE) cells were purchased from ATCC (PCS-400–010) and cultured in RenaLife Basal Medium with supplements (Lifeline Cell Technology LL-0025) as previously described (*Abend et al., 2007*). COS-7 cells were purchased from ATCC (CRL-1651) and cultured in DMEM medium (Corning Cellgro 10–017-CV) supplemented with 5% fetal bovine serum (FBS) (Seradigm). HEK-293 cells were purchased from ATCC (CRL-1573) and cultured in DMEM medium (Corning Cellgro 10–017-CV) supplemented with 10% FBS (Seradigm). Cells were cultured at 37°C with 5% $CO_2$.

### Virus stock generation

BKV stocks were generated by transfection and infection of cells as previously described (*Abend et al., 2007*). Briefly, BKV ST1 MM viral genome was excised from pBR322 plasmid (ATCC 45026) using *BamHI* (NEB), cleaned up using QIAquick PCR Purification Kit (Qiagen), and re-circularized using T4 DNA ligase (NEB) overnight at 16°C. The re-ligated viral genomes were extracted using phenol:chloroform:isoamyl alcohol (25:24:1,v/v) (Sigma) and aqueous phase was separated using Phase Lock Gel Heavy tube (5Prime), followed by ethanol precipitation and resuspension of viral genomes in Buffer EB (10 mM Tris-HCl, pH 8.5, Qiagen). HEK-293 cells were transfected with 2–4 µg of viral genome using Lipofectamine 2000 (Invitrogen) and Opti-MEM (Gibco) according to manufacturer's protocol, and cells were cultured for 10–14 days until cytopathic effects (CPE) were observed. Cells were freeze-thawed three times and supernatant cleared by centrifugation at 1600 rpm for 15 min. Low-titer virus from resulting supernatant was used to infect either RPTE or HEK-293 cells, and cells were cultured for 12–14 days (RPTE cells) or 21–28 days (HEK-293 cells) until CPE was observed. Cells were then scrapped, freeze-thawed three times, and purified as described below.

JCV stocks were prepared similarly. The genome of JCV genotype Ia isolate Mad-1 (GenBank Accession J02227) cloned into the pBR322 plasmid at the *EcoRI* restriction site (resulting construct: pM1TC) was a generous gift from Walter Atwood (Brown University). JCV stocks were produced by transfection and infection of cells similar to what has been previously described (*Hara et al., 1998*). Briefly, the viral genome was first extracted from the plasmid backbone by digestion with *EcoRI* (NEB) and re-circularized using T4 DNA ligase. The resulting viral genomes were then purified using the QIAquick PCR Purification Kit (Qiagen) to prepare for transfection into cells. COS-7 cells, a cell line supportive of JCV replication (*Hara et al., 1998*), were seeded 1 × 10^6 cells per T75 flask and transfected with 2–4 µg of viral genomes using Lipofectamine 2000 (Invitrogen) and Opti-MEM (Gibco) according to manufacturer's protocol. Transfected cells were incubated at 37°C with 5% $CO_2$ for 4 hr, then transfection medium was replaced with infection medium (DMEM supplemented with 2% FBS and 1X Pen/Strep). Cells were cultured at 37°C with 5% $CO_2$ for 6–10 weeks, until (CPE) became evident. During this time, 2–3 mL fresh infection medium was added every 3–4 days and cells were split by 1:2 to 1:3 dilution factors once a week to maintain cell health and prevent overcrowding. Upon observation of significant CPE, cells were collected by scraping, combined with culture media, subjected to three freeze-thaw cycles to release intracellular virus. These resulting viral stocks were titrated and stored at −80°C.

### Virus purification

Purified BKV was prepared as previously described (*Jiang et al., 2009*). Briefly, crude lysate containing high-titer BKV was cleared by centrifugation at 3200 rpm for 30 min at 4°C, and supernatant (S1) was separated from the resulting pellet. The pellet (P1) was resuspended in buffer A (10 mM HEPES, pH 7.9, 1 mM CaCl$_2$, 1 mM MgCl$_2$, 5 mM KCl). The resuspended pellet pH was lowered to 6.0 with 0.5 M HEPES (pH 5.4), and incubated with neuraminidase (1 U/mL; Sigma) for 1 hr at 37°C. Pellet buffer pH was then raised pH 7.4 with 0.5 M HEPES (pH 8), and cleared by centrifugation at 16,000 x g for 5 min at 4°C. The resulting supernatant (S2) was pooled with the initial (S1), and the pellet (P2) was resuspended in buffer A containing 0.1% deoxycholate (Sigma), incubated for 15 min at room temperature, cleared by centrifugation at 16,000 x g for 5 min at 4°C, and the resulting supernatant (S3) was pooled with the other supernatant fractions. Pooled supernatants were placed over

a 4 mL 20% (w/v) sucrose solution and centrifuged at 83,000 x g for 2 hr at 4°C in a SW32Ti rotor (Beckman). The resulting pellet was resupended in 2 mL buffer A, and placed over a CsCl gradient from 1.2 to 1.4 g/cm$^3$ in buffer A generated using a J17 gradient former (Jule, Inc), and centrifuged at 35,000 rpm for 16 hr at 4°C in an SW41 rotor (Beckman). The BKV band formed in the gradient was collected using an 18-guage needle, and dialyzed in a Slide-A-Lyzer Dialysis Cassette, 10K MWCO (ThermoFisher Scientific) over 2 days in 2L buffer A at 4°C, with buffer exchanged once during dialysis. BKV was then aliquoted and stored at −80°C.

## Antibodies and reagents

The following primary antibodies were used in this study: monoclonal mouse anti-SV40 T-antigen (PAb416, EMD Millipore;) at 1:200 for immunofluorescent staining (IF), monoclonal mouse anti-BKV VP1 antibody (in-house generated) at 1:500 for IF, polyclonal rabbit anti-SV40 VP1 (Abcam) at 1:500 for IF and 1:1000 for immunoblotting (IB), polyclonal rabbit anti-SV40 VP2/3 (Abcam) at 1:1000 for IF and 1:1000 for IB, polyclonal rabbit anti-BiP (Abcam) at 1:1000 for IB, monoclonal mouse anti-HSP90 (Abcam) at 1:1000 for IB, and polyclonal rabbit anti-biotin (Abcam) at 1:750 for IF. The following secondary antibodies were used in this study: in IF applications, goat anti-mouse IgG conjugated to either Alexa Fluor 488, 594, or 647 (ThermoFisher Scientific), goat anti-rabbit IgG conjugated to either Alexa Fluor 488 or 594 (ThermoFisher Scientific); in IB applications, goat anti-mouse IgG conjugated to IRDye 680RD (Li-COR), goat anti-rabbit IgG conjugated to IRDye 800CW (Li-COR). The human anti-BKV VP1 IgG1 antibody P8D11 was produced by the Novartis Institutes for BioMedical Research Biologics Center. D1$_{min}$, D1$_{min}$ W293A, D1$_{min}$ L297A, D1$_{min}$ Y302A, and biotin-peptide probe for the biochemical assay were synthesized and HPLC-purified by the Tufts University Core Facility with purity ≥90%. TAT-fused D1$_{min}$ peptides were synthesized by CPC Scientific. All other peptides were synthesized by the Sigma Chemical Company, with purity determined by LCMS to be 35–74%.

## Immunofluorescent staining

For T-antigen staining, cells were fixed with 4% paraformaldehyde (w/v) in PBS for 15 min, then incubated with primary antibody in 0.2% gelatin, 0.1% Triton X-100 in PBS for 1 hr, followed by incubated with secondary antibody at 1:3000 and 4′,6-diamidino-2-phenylindole (DAPI, Calbiochem) contrast stain at 1.67 µg/mL in 0.2% gelatin in PBS for 1 hr. For VP1 co-localization and cell-binding assays, cells were fixed with 4% paraformaldehyde (w/v) in PBS for 15 min then permeabilized with 0.1% Triton X-100 in PBS for 10 min. Cells were then blocked with 2% goat serum (Invitrogen) for 30 min, then incubated with primary antibodies for 1 hr, and secondary antibodies for 1 hr, followed by a 10 min incubation with DAPI contrast stain at 1.67 µg/mL (Calbiochem). For VP2/3 staining, cells were fixed in 100% methanol for 15 min at −20°C then blocked in 3% nonfat milk (Bio-Rad), 0.1% Tween-20 (Bio-Rad) in PBS for 30 min. Cells were then incubated with anti-VP1 and anti-VP2/3 antibodies for 1 hr, and secondary antibodies for 1 hr, followed by a 10 min incubation with DAPI contrast stain at 1.67 µg/mL.

## Infections

Viral titers were measured by fluorescent focus assay, as previously described (*Jiang et al., 2009*). For 96-well plate format assays, RPTE cells were seeded 12,000 per well. For ER-to-cytosol retro-translocation assays, RPTE cells were seeded in 6-well plates at 380,000 cells per well. For non-synchronized infections of RPTE cells, virus was diluted in RenaLife medium and added to cells followed by incubation at 37°C for the desired time. For synchronized infections, cells were pre-chilled to 4°C for 15 min. Purified virus was diluted into cold RenaLife medium and incubated with cells for 1 hr at 4°C. Cells were rinsed once with cold RenaLife medium, followed by addition of warm medium and incubation at 37°C for the desired time. For COS-7 cell assays, cells were seeded 5,000 per well in a 96-well plate format. COS-7 cells were infected using a synchronized infection protocol as described above, with the following modifications: JCV or BKV were diluted into low-serum medium (DMEM supplemented with 2% FBS), and cells were rinsed with cold low-serum medium and cultured in low-serum medium at 37°C for the desired time.

Spreading infection assays were performed as follows. Re-circularized BKV genomes were prepared as described. RPTE cells were reverse-transfected with 100 ng viral genome DNA using

Lipofectamine 3000 (Invitrogen) at a 1.5:1 ratio of L3000 to DNA, and Opti-MEM (Gibco) in a 96 well-plate format. Medium was exchanged the following day, and plates were incubated at 37°C for the desired time.

## Preparation of BKV mutants

Mutant BKV genomes were generated by PCR-based site-directed mutagenesis using the primers listed below.

| Name | Forward primer (5′-to-3′) | Reverse primer (5′-to-3′) |
|---|---|---|
| VP1 P232S | caggaggggaaaatgttTCCccagtacttcat | cacatgaagtactggGGAaacattttcccctcctg |
| VP1 P232L | caggaggggaaaatgttCTCCccagtacttcat | cacatgaagtactggGAGaacattttcccctcctg |
| VP1 P232I | caggaggggaaaatgttATCccagtacttcat | cacatgaagtactggGATaacattttcccctcctg |
| VP1 V234S | gaaaatgttcccccaTCActtcatgtgaccaac | gtgttggtcacatgaagTGAtgggggaacatt |
| VP1 V234L | gaaaatgttcccccaTTActtcatgtgaccaac | gtgttggtcacatgaagTAAtgggggaacatt |
| VP1 V234I | gaaaatgttcccccaATActtcatgtgaccaac | gtgttggtcacatgaagTATtgggggaacatt |
| ΔVP2 | gtatttccaggttcatAggtgctgctctagcacttttgggggac | gagcagcaccTatgaacctggaaatacaaaaaaaaagggattac |
| ΔVP3 | gcaatcaggcatAgctttggaattgtttaacccagatgagtac | ccaaagcTatgcctgattgctgatagaggcctacagtggaaac |
| VP2 P291A | caaagaactgctGctcaatggatgttgcctttacttctaggcc | catccattgagCagcagttctttgattagcacctcctgg |
| VP2 W293A | ctgctcctcaaGCgatgttgcctttacttctaggcctgtac | ggcaacatcGCttgaggagcagttctttgattagcacctcc |
| VP2 L297A | gatgttgcctGCacttctaggcctgtacgggactgtaacac | caggcctagaagtGCaggcaacatccattgaggagcagttc |
| VP2 Y302A | ctaggcctgGCcgggactgtaacacctgctcttgaagcatg | gttacagtcccgGCcaggcctagaagtaaaggcaacatccattg |

After PCR, reactions were treated with *DpnI* (NEB) to remove template DNA, and PCR products were used to transform XL10-Gold (VP1 mutants, Agilent) or 10-beta cells (VP2 mutants, NEB). Resultant colonies were sequenced and analyzed for desired mutation, and viral genomes were prepared as described. Point mutations lead to amino acid substitutions in both VP2 and VP3. Deletion mutants were obtained by point mutation of the start codon. While ΔVP2 did not affect VP3 sequence, ΔVP3 resulted in a M120I substitution in VP2. BKV ΔVP2ΔVP3 genome was generated by successive rounds of site-directed mutagenesis with ΔVP2 and ΔVP3 primer sets.

## Inhibitor treatments

Dose-response curves were determined using 3-fold, 10-point titrations of inhibitor. For peptide $EC_{50}$ determination, cells were treated with inhibitors for two hours prior to infection. For synchronized infection treatments, inhibitors were added immediately following synchronization; $EC_{50}$ concentration for P8D11 was determined using the synchronized treatment protocol. $CC_{50}$ values were determined using a CellTiter-Glo luminescent cell viability assay (Promega) after two days of treatment, with luminescence detected on a PHERAstar FS (BMG Labtech).

## Peptide-virus co-purification assay

54 nM peptide was incubated with 75 μg/mL (370 nM) recombinant full-length VP1 pentamers, or molar equivalent of VP1 pentamers of BK VLP or purified infectious BKV (as assessed by SDS-PAGE and comparing VP1 abundance by Coomassie stain) in 336 μl PBS buffer containing 1% DMSO and 0.01% Tween-20 for 1 hr at room temperature. 300 μl of mix was added to a fresh tube, and 20 μl of Dynabeads M-280 Streptavidin (Invitrogen) were added. Samples were rotated for 1 hr at room temperature, then rinsed three times with 1 mL wash buffer (PBS, 1% DMSO, 0.01% Tween-20), using a DynaMag-2 Magnet (Invitrogen) to separate beads from solution. After washes, beads were incubated in 20 μl elution buffer (PBS, 0.1% SDS) and denatured at 95°C for 5 min. Eluate was removed from beads, and input and eluate samples were mixed with 4X LDS sample buffer with reducing agent (Invitrogen), heated at 70°C for 10 min, then samples were analyzed by SDS-PAGE and visualized using InstantBlue Coomassie staining reagent (Expedeon).

## BKV cell-binding assay

Purified infectious BKV was incubated with titrated concentrations of $D1_{min}$ peptide or P8D11 antibody for 1 hr at 4°C in RenaLife medium. RPTE cells seeded in 96-well plate format were cooled for 15 min at 4°C, and medium containing virus and inhibitor mix was added to cells and incubated for 1 hr at 4°C. Cells were rinsed with cold RenaLife medium, and proceeded to fixation and staining as described.

## Time-of-addition assay

RPTE cells seeded in 96-well plate format were subjected to a synchronized infection at low MOI (MOI = 0.3). Timepoint 0 h.p.i. samples were treated with inhibitor compounds immediately after synchronized infection, others added at the timepoint indicated. Inhibitors were used at the following concentrations: $D1_{min}$, $D1_{min}$-TAT, TAT-$D1_{min}$: 0.8 µM; P8D11: 0.014 µM (2 µg/mL). Plates were incubated at 37°C and fixed at 48 h.p.i.

## Fractionation assay

RPTE cells were cultured in 2 wells in 6-well format per treatment. Cells were subjected to synchronized infection at high MOI (MOI = 10), and treated with 5 µM peptide immediately after synchronized infection. Cells were harvested 24 h.p.i. with 0.05% trypsin, 0.02% EDTA (Lifeline Cell Technology) for 2 min until cells were detached. Trypsin was inhibited with an equal volume of Trypsin Neutralizing Solution (Lifeline Cell Technology), and wells rinsed with phosphate-buffered saline (PBS). Cells were pelleted at 90 x g for 5 min at 4°C, and rinsed with 1 mL cold PBS buffer. Cell pellets were then lysed in 50 µl HNF buffer (150 mM HEPES pH 7.2, 50 mM NaCl, 2 mM $CaCl_2$) containing 0.025% digitonin (Thermo Lifesciences) and 1X cOmplete, Mini Protease Inhibitor Cocktail (Roche) for 10 min on ice. Lysates were then clarified with a 21,100 x g centrifugation at 4°C for 10 min. The supernatant (cytosolic fraction) was removed and the pellet was rinsed with 1 mL HNF buffer and transferred to a fresh tube, and pelleted again at 21,100 x g centrifugation at 4°C for 10 min. Pellets were then resuspended in sample buffer directly in sample buffer. Samples were boiled at 95°C for 10 min then stored at −20°C until subjected to SDS-PAGE and immunoblotting.

## Immunoblotting

Samples were prepared in sample buffer (NuPAGE LDS Sample Buffer (Invitrogen), 100 mM dithiothreitol (DTT)), boiled for 5 min at 95°C, and subjected to SDS-PAGE using 4–12% Bis-Tris Bolt gels (Invitrogen) in MOPS running buffer (Invitrogen). Proteins were transferred to Immobilon-FL PVDF membrane (Millipore) in transfer buffer containing 1X NuPAGE transfer buffer (Invitrogen), 20% methanol, and 0.05% SDS for 70 min. Membranes were blocked using Odyssey blocking buffer (TBS) (Li-COR) for 30 min at room temperature, and incubated with primary antibody diluted in Odyssey blocking buffer containing 0.05% Tween-20 overnight at 4°C. Membranes were rinsed three times with Tris-buffered saline (TBS) containing 50 mM Tris-HCl, pH 7.4, 150 mM NaCl, and 0.1% Tween 20 (TBS-T), and incubated with secondary antibodies for 1 hr at room temperature. Membranes were again rinsed three times with TBS-T and once with TBS containing no detergent, and membranes were immediately scanned using an Odyssey Infrared Imager (Li-COR).

## Imaging and image segmentation

Images were acquired on an ImageXpress Micro XLS system (Molecular Devices) with a 10x objective for assays determining percent infected cells (Nikon CFI Plan Fluor, MRH00101), or a 60x objective for all other assays (Nikon 60X Plan Apo λ, MRD00605). Images were processed using CellProfiler version 2.1.2 (*Kamentsky et al., 2011*).

## Data analysis

$EC_{50}$ values were calculated using XLFit v5.5.0.5 (IDBS). Quantification and processing of data generated by CellProfiler was performed using R v3.5.1 (*R Development Core Team, 2018*). Microscopy data were quantified per field of view, averaged per well, and data displayed as the mean value ± SD across replicate wells. Immunoblot images were processed using Fiji built on ImageJ v1.52b (*Schindelin et al., 2012*).

## VP1 plasmid construction

Synthetic DNA, codon optimized for Sf9 cell expression, encoding full-length BKV serotype 1 VP1 and JCV VP1 were generated for expression of VP1 proteins. For VLP production, DNA fragments encoding full length VP1 were inserted into the pFastBac1 plasmid and baculovirus was generated following the Bac-to-Bac method (Invitrogen). For VP1 pentamer production, DNA fragments encoding either BKV VP1 residues (2-362), JCV VP1 residues (2-354), or BKV VP1 residues (30-297) were inserted into a gateway adapted pGEX plasmid for expression in *E. coli* with an *N*-terminal GST-6xHis-Tev tag. The mutations P232S, V234S, and T224A/T243A were introduced into the BKV $VP1_{30-297}$ plasmid by QuikChange site-directed mutagenesis.

## VP1 pentamer production

BL21 Star (DE3) *E. coli* cells were transformed with expression plasmids and plated on LB agar plates supplemented with 100 μg/ml carbenicillin. Cell were grown in Terrific broth (supplemented with 15 mM sodium phosphate pH 7.0, 2 mM $MgCl_2$, 100 μg/ml carbenicillin) with shaking at 37°C until the OD600 reached 0.7, the temperature was then lowered to 18°C, and Isopropyl β-D-1 thiogalactopyranoside (IPTG) added to 0.5 mM. After 16 hr, harvested cells by centrifugation and stored at −80°C. Cells were re-suspended in chilled lysis buffer (25 mM Tris-HCl pH 8.0, 200 mM NaCl, 5% glycerol, 1 mM Tris(2-carboxyethyl) phosphine (TCEP), 15 mM Imidazole, 1X Roche complete EDTA-free protease inhibitor cocktail, 1X Pierce universal nuclease) at a ratio of 5 mL buffer per gram of cell paste. Cells were lysed by passing through an M-110P microfluidizer at 17,500 PSI on ice. The lysate was centrifuged at 26,800 x g for 45 min at 4°C. 10 mL Nickel sepharose Fast Flow (GE) column was equilibrated in lysis buffer and clarified lysate loaded . Resin was washed with three column volumes (CV) of lysis buffer, then washed with 5 CV of wash buffer (25 mM Tris-HCl pH 8.0, 200 mM NaCl, 1 mM TCEP, 40 mM Imidazole, 5% glycerol) and VP1 was eluted with 2 CV elution buffer (25 mM Tris-HCl pH 8.0, 200 mM NaCl, 1 mM TCEP, 250 mM Imidazole, 5% glycerol). The *N*-terminal tag was removed by cleavage with Tev protease and pentamers were loaded onto a Superdex 200 column equilibrated in SEC buffer (25 mM Tris-HCl pH 8.0, 100 mM NaCl, 1 mM TCEP, 5% glycerol). Peak fractions were pooled then concentrated using a 50,000 molecular weight cut- off (MWCO) Amicon concentrator.

## Labeled VP1 pentamer production

BL21 Star (DE3) *E. coli* cells transformed with VP1 expression plasmid were grown in M9 minimal media (6 g/L $Na_2HPO_4$, 3 g/L $KH_2PO_4$, 0.5 g/L NaCl, 2 mM $MgSO_4$, 0.1X Vitamins (Sigma R7256), 1 g/L $^{15}NH_4Cl$, 3 g/L glucose, 100 μg/mL carbenicillin, 0.1X trace metals) prepared in $D_2O$. Cells were incubated at 37°C with shaking at 250 RPM until the OD600 reached ~0.7. To each liter of culture 70 mg of 2-Ketobutyric acid-4-$^{13}$C,3,3-d2 sodium salt hydrate, 120 mg of 2-Keto-3-methyl-$^{13}$C-butyric-4-$^{13}$C, 3-d acid sodium salt, 100 mg deuterated glycine, and 100 mg L-threonine (4-$^{13}$C;2,3-D2) was added. After 1 hr, the temperature was lowered to 24°C and IPTG was added to 250 μM. After 16 hr, cells were harvested by centrifugation and stored at −80°C. VP1 pentamers were purified as described above with the exception of the SEC buffer being 50 mM Na-Phosphate pH 7.0, 100 mM NaCl.

## VLP production

Recombinant baculovirus encoding untagged full length BKV VP1 or JCV VP1 were used to infect Sf9 insect cells in suspension at $1.5 \times 10^6$ cells/mL, the cells were incubated at 27°C with shaking at 120 RPM for 72 hr then harvested by centrifugation and stored at −80°C. Cells were re-suspended in lysis buffer (20 mM Tris-HCl pH 7.5, 1 M NaCl, 1X Roche EDTA free protease inhibitor cocktail) at a ratio of 10 mL lysis buffer per gram of cell pellet and lysed by sonication on ice then the lysate was centrifuged at 16,000 x g for 20 min at 4°C. The supernatant was layered onto 3 mL of 40% glucose made up in 1X PBS and centrifuged at 116,000 x g for 2.5 hr at 4°C. Dissolved pellet in IEX buffer A (25 mM Tris-HCl pH 8.0, 25 mM NaCl) and loaded onto a 10 mL Sepharose Q-HP (GE) column equilibrated in IEX buffer A. Column was washed with 3 CV IEX buffer A and eluted with a linear NaCl gradient from 25 mM to 700 mM NaCl across 25 CV. Pooled peak fractions and loaded onto 10 mL Capto Core 700 (GE) resin equilibrated in SEC buffer (25 mM Tris-HCl pH 8.0, 100 mM NaCl)

collecting the flow-through fraction. Loaded onto a Sephacryl S500 26/60 column (GE) and collected peak fractions, concentrated with a 100,000 MWCO Amicon concentrator.

## Protein reagents

The following VP1 sequence was used for the biochemical AlphaScreen assays:

BKV serotype 1 VP1 (30–297):
    GKGGVEVLEVKTGVDAITEVECFLNPEMGDPDENLRGFSLKLSAENDFSSDSPERKMLPCYSTARI
PLPNLNEDLTCGNLLMWEAVTVQTEVIGITSMLNLHAGSQKVHEHGGGKPIQGSNFHFFAVGGDPLE
MQGVLMNYRTKYPEGTITPKNPTAQSQVMNTDHKAYLDKNNAYPVECWIPDPSRNENTRYFGTFTGGE
NVPPVLHVTNTATTVLLDEQGVGPLCKADSLYVSAADICGLFTNSSGTQQWRGLARYFKIRLRKRSVK

    The following VP1 sequence was used for SPR assays:

BKV serotype 1 VP1 (2–362):
    GAPTKRKGECPGAAPKKPKEPVQVPKLLIKGGVEVLEVKTGVDAITEVECFLNPEMGDPDENLRGFSLK
LSAENDFSSDSPERKMLPCYSTARIPLPNLNEDLTCGNLLMWEAVTVQTEVIGITSMLNLHAGSQK
VHEHGGGKPIQGSNFHFFAVGGDPLEMQGVLMNYRTKYPEGTITPKNPTAQSQVMNTDHKAYLDKNNA
YPVECWIPDPSRNENTRYFGTFTGGENVPPVLHVTNTATTVLLDEQGVGPLCKADSLYVSAADICGLF
TNSSGTQQWRGLARYFKIRLRKRSVKNPYPISFLLSDLINRRTQRVDGQPMYGMESQVEEVRVFDGTE
RLPGDPDMIRYIDKQGQLQTKML

## Chemical attachment of biotin to VP1 proteins

For SPR analysis, biotin was covalently attached to BKV $VP1_{2-362}$ with the sulfo-N-hydroxysuccinimide (NHS) ester of a biotin derivative (ThermoFisher Scientific # 21338) as follows: to a 1500 µl of a solution of BKV $VP1_{2-362}$ protein at 17 µM in PBS buffer containing 1 mM TCEP was added 8 µl of a 1 mg/mL (1.5 mM) solution of sulfo-NHS-LC-LC-biotin in water. The solution was mixed with a vortex mixer briefly (1 s), and incubated at room temperature for 1 hr. The solution was transferred to a ThermoFisher Slide-A-Lyzer dialysis cassette (3.5 kDa MWCO, 3 mL) and dialyzed extensively against 3 times 2L of PBS buffer containing 1 mM TCEP at 4°C for 18 hr.

## Analysis of peptide:VP1 interactions by surface plasmon resonance

SPR analysis for the determination of the dissociation constant $K_D$ was performed with a Biacore T200 instrument with PBS buffer containing 1 mM TCEP, 0.05% Tween-20 (or P20, GE Healthcare) and 1 mM ethylenediaminetetraacetic acid (EDTA) at 20°C. The flow rate was 60 µl per minute. $VP1_{2-362}$ protein covalently modified with biotin was loaded onto a streptavidin-coated Biacore Series S Sensor Chip SA biosensor (GE Healthcare) that had been pre-treated with 50 mM NaOH containing 1 M NaCl. The protein loading response was 6000–8000 resonance units. Peptides were analyzed using the single cycle kinetics method according to instrument control software instructions. Data were analyzed using Biacore Evaluation Software to generate affinity constants ($K_D$).

## AlphaScreen competitive binding assay

The assay was run in a Tris buffer at pH 7.5 containing 100 mM NaCl, 0.01% Tween-20, 1 mM EDTA and 0.01% bovine serum albumin. The $D1_{22}$ biotin-peptide probe with the sequence [H]-APGGA NQRTAPQWMLPLLLGLYG-GGGK(Biotin)-[OH] was incubated with BKV $VP1_{30-297}$ for 90 min before addition of an anti-BKV VP1 antibody (in-house generated) along with AlphaScreen streptavidin donor and protein A acceptor beads (PerkinElmer). Samples were incubated overnight before reading on a PerkinElmer Envision. Untagged peptides were assessed in a competition mode where they were serially diluted in assay buffer and added to the VP1 along with the biotin-peptide probe. Peptide $IC_{50}$ values (n = 3) were calculated in Microsoft Excel using XLfit.

## Co-crystallization of BKV VP1 pentamer with $D1_{min}$

For co-crystallization, 13-mer $D1_{min}$ peptide was added to BKV VP1 protein to a final concentration ratio of 5:1 $D1_{min}$ to pentamer. The resultant mixture was incubated on ice for 1 hr then concentrated to 15 mg/mL protein overall. Prior to crystallization, the mixture was passed through a 0.2-micron filter. The protein-peptide complex was crystallized using the hanging drop vapor diffusion

method. 2.0 μL of protein solution was mixed with 2.0 μL of well solution (20% PEG-3350, 5% ethylene glycol, 0.1 M Tris buffer pH 8.5, 10 mM TCEP). The resulting drop was suspended over a reservoir of 0.3 mL well solution. The crystals grew at 18°C for approximately 12–24 hr. Crystals were washed briefly in a cryoprotectant consisting of 80% well solution and 20% ethylene glycol (v/v) and then flash-frozen in liquid nitrogen prior to data collection.

## Structure solution and refinement of BKV VP1 pentamer:D1$_{min}$ complex

The X-ray diffraction data were collected at a wavelength of 1.54187 Å and a temperature of 100K on a Rigaku FRE+ anode utilizing a Decris 300K Pilatus detector. Data integration and scaling were performed by using the autoPROC implementation of XDS and AIMLESS (*Vonrhein et al., 2011*). The structure of the complex was solved via Molecular Replacement using the CCP4i Suite implementation of PHASER (*McCoy et al., 2007*; *Winn et al., 2011*). The structure was built and refined via alternating rounds of real-space rebuilding in Coot and refined using autoBuster (Global Phasing) until convergence was reached. Data reduction and structure refinement statistics are presented in *Supplementary file 3*. After attempts to refine the D1$_{min}$ model to convergence, with suitable φ/ψ angles as defined by the Ramachandran plot and suitable rotamers, it became clear that a single binding model could not account for the electron density seen in 2Fo-Fc maps. A model that rationalized the electron density was achieved by fitting the peptide with an occupancy of 0.8. Subsequent reciprocal space and real-space refinement of this model led to a suitable fit of the peptide into the observed density and reduced, but still unaccounted for, difference density for the remaining symmetry-related binding modes, which are not fit. As such, the co-structure is presented as a model constructed using the observed density, or a 'structurally-guided model.' Alignment of structurally-guided model with apo BKV pentamer X-ray structure (PDB: 4MJ1; *Neu et al., 2013*) was performed using the align tool in PyMOL v2.2.3 (Schrödinger, LLC) with a 10 Å cutoff for outliers. Post-alignment, RMSD values were calculated using the tool rms_cur without refitting the alignment.

## NMR spectroscopy

All NMR experiments were performed on a Bruker Avance III 600 MHz spectrometer equipped with a 5 mm z-gradient QCI-F cryo probe. The temperature in all experiments was 32°C (305K). The NMR samples were prepared in 160 μL PBS buffer at pH 7.5, containing 2 mM deuterated DTT, 10% (v/v) D$_2$O and 11.1 μM 4,4-dimethyl-4-silapentane-1-sulfonic acid (DSS, internal standard). The final protein concentration of truncated (residues 30–297), $^2$H,$^{12}$C,$^{15}$N and $^1$H,$^{13}$C-methyl-ILVT labeled BKV VP1 was 125 μM (monomer concentration) in all experiments. The 13-mer D1$_{min}$ (Ac-APQWMLPLLLGLY-NH$_2$) and alanine-substitution peptide D1$_{min}$ W293A (Ac-APQAMLPLLLGLY-NH$_2$) were dissolved in d$_6$-DMSO and added to the protein at various concentrations (6.25 μM – 200 μM).

1D-$^1$H NMR experiments were acquired with 64 scans, excitation sculpting water suppression and a relaxation delay of 2 s. 2D $^1$H,$^{13}$C-HMQC SOFAST (*Schanda et al., 2005*) spectra were recorded using 50% non-uniform sampling, 1024 and 256 points in the direct and indirect dimensions, respectively, 192 scans and a recycling delay of 200 ms. Spectra were processed and analyzed using TOPSPIN version 3.5. Methyl peak assignments were obtained by means of twelve amino acid point mutations (L68A, L254M, V136I, V231I, V234I, T46S, T118S, T224S, T238S, T240S, T277S, I45V; see *Figure 2—figure supplement 1.B* for an example of this method) and by using a methyl walk approach based on $^{13}$C-resolved 4D-HMQC-NOESY-HMCQ experiments (*Proudfoot et al., 2016*).

## MOI dependency assay

RPTE cells seeded in 96-well plate format were cooled for 15 min at 4°C, then cells were treated with purified infectious BKV at either low (MOI = 0.3) or high (MOI = 5) viral titers. After 1 hr, cells were rinsed with cold RenaLife medium and warm RenaLife medium containing DMSO, or 5 μM D1$_{min}$ WT or L297A peptide was added to cells. Plates were incubated at 37°C and fixed at 48 h.p.i. as described. T-antigen staining and EC$_{50}$ determination were performed as described.

## Real-time quantitative PCR

RPTE cells seeded in 6-well format were treated with DMSO, 5 μM D1$_{min}$ WT or L297A peptide, or 1000 U/mL of IFN-β (Peprotech) for 20 hr at 37°C, then harvested by incubation with 0.05% trypsin, 0.02% EDTA (Lifeline Cell Technology) for 2 min until cells were detached. Trypsin was inhibited with

an equal volume of Trypsin Neutralizing Solution (Lifeline Cell Technology), and wells rinsed with PBS. Cells were pelleted at 124 x g for 5 min at 4°C, and rinsed with 1 mL cold PBS buffer, and pelleted again at 124 x g for 5 min at 4°C. Cell pellets were frozen and stored at −80°C. Total cell RNA was extracted using an RNeasy mini kit (Qiagen) and cDNA generation was performed with 500 ng total cell RNA using SuperScript IV VILO Master Mix (Invitrogen) using the following incubations: 25°C for 10 min, 50°C for 10 min, 85°C for 5 min. Resultant cDNA was diluted 1:5 in water and 2 µl was run in triplicate in 10 µl reactions with Fast SYBR Green Master Mix (Applied Biosystems) in 384-well format with 500 nM of the following PrimeTime qPCR primers (IDT) or water:

| Gene | PrimeTime Primer |
| --- | --- |
| GAPDH | Hs.PT.39a.22214836 |
| CXCL10 | Hs.PT.58.3790956.g |
| IFNA2 | Hs.PT.58.24294810.g |
| IFNB1 | Hs.PT.58.39481063.g |
| MX1 | Hs.PT.58.40261042 |
| OAS1 | Hs.PT.58.2338899 |
| STAT1 | Hs.PT.58.15049687 |

qPCR was performed on a 7900HT Fast Real-Time PCR System (Applied Biosystems) using the following protocol: 1 cycle at 95°C for 20 s, then 40 cycles of 95°C for 3 s then 60°C for 30 s. A threshold of 0.1 was applied, $\Delta\Delta C_T$ values were calculated normalized to GAPDH and relative to DMSO control using mean triplicate $C_T$ values, and relative fold-change was calculated as $2^{-\Delta\Delta CT}$.

### Peptide cell-binding assay

RPTE cells seeded in 96-well plate were incubated with 1 µM $N$-terminal biotinylated $D1_{22}$, 5 µM $D1_{min}$ WT or L297A, and/or 10 nM BK VLP for 1 hr at 37°C. Cells were rinsed with PBS then proceeded to fixation and staining as described, except the permeabilization step with Triton X-100 was skipped to avoid staining intracellular biotin.

## Acknowledgements

We thank Weidong Zhong, Kelly Wong, and Don Ganem for their leadership roles, Catherine Jones for important intellectual contributions in project team management, Atul Sathe, Lihong Zhao, and Sue Ma for their guidance, advice, and assistance with polyomavirus virology, and members of the local postdoctoral community for their support. All authors were funded by Novartis Institutes for Biomedical Research. JRK is a postdoctoral fellow at Novartis Institutes for Biomedical Research.

## Additional information

#### Competing interests

Joshua R Kane, Susan Fong, Jacob Shaul, Alexandra Frommlet, Andreas O Frank, Mark Knapp, Dirksen E Bussiere, Peter Kim, Elizabeth Ornelas, Carlos Cuellar, Anastasia Hyrina, Johanna R Abend, Charles A Wartchow: was employed by the Novartis Institutes for BioMedical Research while this work was conducted.

## Funding

| Funder | Grant reference number | Author |
| --- | --- | --- |
| Novartis | | Joshua R Kane<br>Susan Fong<br>Jacob Shaul<br>Alexandra Frommlet<br>Andreas O Frank<br>Mark Knapp<br>Dirksen E Bussiere<br>Peter Kim<br>Elizabeth Ornelas<br>Carlos Cuellar<br>Anastasia Hyrina<br>Johanna R Abend<br>Charles A Wartchow |
| Novartis | Postdoctoral fellowship | Joshua R Kane |

The funders had no role in study design, data collection and interpretation, or the decision to submit the work for publication.

## Author contributions

Joshua R Kane, Investigation, Visualization, Methodology, Wrote the manuscript, Performed peptide antiviral assays, peptide cytotoxicity assays, in vitro peptide affinity purifications, fractionation assays, VP2/3 exposure studies, peptide-cell binding assay, and associated data analysis, Performed time-of- addition study, Performed mutant virus spread assays, Performed RT-qPCR study; Susan Fong, Investigation, Methodology, Performed biophysical and biochemical assays; Jacob Shaul, Resources, Methodology, Performed purification of protein reagents and VLPs; Alexandra Frommlet, Andreas O Frank, Investigation, Visualization, Methodology, Performed NMR experiments; Mark Knapp, Investigation, Visualization, Methodology, Performed X-ray crystallography studies; Dirksen E Bussiere, Investigation, Methodology, Performed X-ray crystallography studies; Peter Kim, Investigation, Visualization, Performed mutant virus spread assays; Elizabeth Ornelas, Carlos Cuellar, Investigation, Performed X-ray crystallography studies; Anastasia Hyrina, Methodology, Performed RT-qPCR study; Johanna R Abend, Conceptualization, Supervision, Investigation, Wrote the manuscript, Performed time- of- addition study; Charles A Wartchow, Conceptualization, Data curation, Supervision, Investigation, Visualization, Methodology, Project administration, Wrote the manuscript, Performed biophysical and biochemical assays

## Author ORCIDs

Joshua R Kane https://orcid.org/0000-0002-6547-638X
Charles A Wartchow https://orcid.org/0000-0002-0123-1455

## Decision letter and Author response

Decision letter https://doi.org/10.7554/eLife.50722.sa1
Author response https://doi.org/10.7554/eLife.50722.sa2

# Additional files

## Supplementary files

- Supplementary file 1. JCV VP1 peptide binding data.
- Supplementary file 2. Additional peptide information.
- Supplementary file 3. Data collection and refinement statistics (molecular replacement).
- Supplementary file 4. Control inhibitor half-maximal effective concentrations ($EC_{50}$) and half-maximal cytotoxic concentrations ($CC_{50}$).
- Supplementary file 5. Key resources table.

## Data availability

All data generated or analysed during this study are included in the manuscript and supporting files. Source data may be found in main and supplemental tables.

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
