## [Decision Letter]

**Acceptance summary:**

Polyomaviruses are important human pathogens and there is a need for anti-polyoma inhibitors for use in immunosuppressed patients, particularly following kidney transplantation. Polyomaviruses enter target cells by endocytosis and then travel to the nucleus via the endoplasmic reticulum. This study shows that a peptide corresponding to 13 residues of the BK polyomavirus minor capsid protein VP2/3 binds tightly to the central pore of the VP1 pentamer on the viral capsid, interferes with an intracellular step that appears to correspond to a viral uncoating transition, and inhibits the infectivity of both BK virus and JC virus. Strengths of the study include the generally high technical quality of the work, the tight binding of the peptide, the novel binding site and mode of action, and the insights on the poorly understood intracellular step(s) that constitute the uncoating transition. The study will be of general interest to molecular virologists, protein biochemists, and those interested in developing therapeutic strategies to combat polyomaviruses.

**Decision letter after peer review:**

Thank you for submitting your article "A polyomavirus peptide binds to the capsid VP1 pore and has potent antiviral activity against BK and JC polyomaviruses" for consideration by *eLife*. Your article has been reviewed by three peer reviewers, and the evaluation has been overseen by a Reviewing Editor and Cynthia Wolberger as the Senior Editor. The following individual involved in review of your submission have agreed to reveal their identity: Daniel DiMaio (Reviewer #2).

The reviewers have discussed the reviews with one another and the Reviewing Editor has drafted this decision to help you prepare a revised submission.

Summary:

Polyomaviruses are important human pathogens that travel to the nucleus via the endoplasmic reticulum. The manuscript by Kane and colleagues shows that a peptide corresponding to 13 residues of the BK virus minor capsid protein VP2/3 binds tightly to VP1 pentamers and inhibits infection of both BK virus and JC virus. Biophysical assays, including NMR and x-ray crystallography, show clearly that the peptide binds in the upper portion of the 5-fold pore formed by 5 VP1 monomers and the binding amino acids were mapped on both the peptide and VP1. Importantly, the peptide can bind whole virions containing both minor capsid proteins so the endogenous VP2/3 protein does not appear to occlude the binding site for this peptide in the mature virion. The data are clear that the binding is within the pore thus making this the first reagent targeting the VP1 pore.

This is an interesting and important study that is largely convincing. Strengths include the generally high technical quality of the work, the tight binding of the peptide, the novel binding site and mode of action indicated by the mapping, the peptide activity against at least one other polyoma virus (JCV), the insights on the poorly understood uncoating transition, and the need for anti-polyoma inhibitors for use in immunosuppressed patients, particularly following kidney transplantation. That said, the study is necessarily mostly descriptive (i.e., showing where the peptide binds, identifying mutations that inhibit binding, and defining the stage of the viral life cycle where inhibition occurs) rather than mechanistic (it isn't clear why blocking this pore binding site inhibits BKV replication), the peptide itself won't be a viable therapeutic (though it might be used to establish a screen for small molecule inhibitors that binding in the same site), and some technical concerns about experimental design and interpretation (see below).

Essential revisions:

1) Authentication of the Peptide Target. There is a possibility that the biological data has been misinterpreted if the peptide is acting as an antagonist to signaling pathways at the cell surface (and the mutants disrupt this interaction). This is easily tested by asking if wild type or mutant peptides bind to the cell and elicit a response (e.g. IFN production). Related issues are that RPTE cells were pre-treated with the peptide for 2 hours before challenging with virus. If the mechanism involves the peptide binding to the virus one would have thought that the virus would have been preincubated with the peptide first and then added to the cells. In that experiment (i.e., preincubation of the virus with peptide first), only cell binding was measured (and was not disrupted), but infection should also be scored.

2) Site of Peptide Activity. Experiments show clearly that endocytosis and delivery to the ER is not affected by the peptide, but subsequent translocation of VP1 to the cytosol is disrupted and VP2 is not exposed in the ER. However, it's unclear from these experiments that the peptide is actually present along with the virus in the ER because the antibody to VP2 recognizes the endogenous VP2. Similarly, it's unclear how and when the peptide would engage the virion given how the experiments were conducted. One has to imagine that the peptide is endocytosed along with the virus (or penetrates the membrane separately – similar to the proposal for the Tat-tagged peptide). What's missing is evidence to show peptide and virus engaged within the cell. Finally, can the time-of-addition experiments be explained by a simple model that the authors have not considered, in which the peptide has to bind to the virion, so that adding it after the virus has entered the cell will be too late? (even if this is true, it does not refute the conclusion that the peptide acts at/before disassembly).

3) Strengthening the Proposed Mechanism of Action. Several simple experiments would significantly strengthen the conclusions.

A) VP1 Release/Virion Disassembly. The Helenius group reported a simple gel-based assay for release of VP1 from the capsid (Cell, 2007). It would be informative to determine if the peptide blocked this step. This is important because the only other evidence for a disassembly block is based on immunofluorescence staining in infected cells, which could be misleading.

B) Peptide Binding and Function. There is a correlation between peptide binding and inhibitory activity, but there are simple experiments that would greatly strengthen this correlation. For example, the VP1 V234L mutation has little effect on infectivity, and the V234I mutation causes an intermediate phenotype. Peptide binding should be checked in these mutants. If binding is defective, the prediction is that the peptide would not inhibit these mutants. Such a result would definitively establish that binding to the mapped site is required for inhibition. Similarly, do the biotinylated peptides inhibit infection? Does only the N-terminal biotinylated peptide inhibit, which is the one that binds? Finally, the correlation between binding affinity to VP1 and inhibitory activity of mutant peptides was established with purified VP1 pentamers. Affinities should be measured with intact virions, which were used in the infectivity assays. Collectively, these experiments would strengthen the correlation between binding and activity.

---

## [Author Response]

Essential revisions:1) Authentication of the Peptide Target. There is a possibility that the biological data has been misinterpreted if the peptide is acting as an antagonist to signaling pathways at the cell surface (and the mutants disrupt this interaction). This is easily tested by asking if wild type or mutant peptides bind to the cell and elicit a response (e.g. IFN production). Related issues are that RPTE cells were pre-treated with the peptide for 2 hours before challenging with virus. If the mechanism involves the peptide binding to the virus one would have thought that the virus would have been preincubated with the peptide first and then added to the cells. In that experiment (i.e., preincubation of the virus with peptide first), only cell binding was measured (and was not disrupted), but infection should also be scored.

A reviewer suggests an alternate explanation for the observed antiviral activity that we indeed had not considered nor discussed, and we thank the reviewer for this contribution. The reviewer-proposed model is an antiviral MOA wherein the peptide binds to an unidentified cell-surface receptor, initiating a signaling cascade that results in the observed antiviral activity associated with D1_min_ treatment. In response, we think that this MOA is unlikely based on the time-of-addition study in the original experiments, which we did not discuss in this context, and based on new additional experimental data in the revised manuscript.

First, the time-of-addition experiments are consistent with VP1 as the target. Treatment with D1_min_ peptides containing the *N-* and *C-*terminal cell-penetrating TAT sequences is consistent with VP1 as the target and not a host receptor. *N-* and *C-*terminal TAT-tagged peptides inhibit biochemical assays with similar IC_50_s as the native peptide (TAT-D1_min_ and D1_min_-TAT 2-3 nM, D1_min_ 4 nM) but they have different profiles in the BKV time-of-addition studies. For the TAT peptides, there is a delay in loss of antiviral activity by >10 hours as compared to 4 hours for the untagged peptide, which is consistent with the cell-penetrating peptide accessing BK VP1 after endocytosis. If the antiviral activity of D1_min_ were due to the binding of the peptide to a cell-surface receptor, the time-of-addition study would show identical profiles for all three peptides since they should bind to the unidentified cell-surface receptor. Since we examined both *C-* and *N-*terminal additions of the TAT sequence to D1_min_, it is unlikely that we are disrupting interactions with an unidentified receptor at either terminus of the peptide.

Next, to further support the hypothesis that the target of the peptide is VP1, we added results to the revised manuscript where we increased the viral titer in an MOI-dependency assay. If the EC_50_ for D1_min_ were observed to be independent of viral titer, the MOA would be consistent with targeting a cell surface receptor (the abundance of which should remain invariant with respect to viral titer), whereas if the EC_50_ were to shift to higher peptide concentrations, this would be consistent with VP1 as the target of the peptide. Consistent with our hypothesis, the EC_50_ shifted 15.4-fold with increased BKV titer in this experiment. The results of these studies are consistent with a mechanism of inhibition that requires the interaction of D1_min_ with viral VP1.

Next, we found that *N*-terminal biotinylated D1_22_ peptide, which contains additional *N*-terminal residues that are not involved in VP1 recognition, does not bind to the cell surface in the absence of VP1. As a positive control for this finding, we observed detection of the biotinylated peptide in the presence of VLPs, which bind to the cell surface and bind the biotinylated peptide.

Lastly, we asked whether treatment with peptide elicited a gene-expression response of interferon-stimulated genes, as suggested by a reviewer. We used RT-qPCR to assess whether genes linked to antiviral activity are induced in RPTE cells after peptide treatment. We observe no measurable induction of genes (CXCL10, IFNA2, IFNB1, MX1, OAS1, and STAT1) with peptide treatment for 20 hours, whereas treatment with interferon-β induced expression of all genes tested (albeit with modest induction of IFNA2 expression). While we did not perform a genome-wide analysis, the results are consistent with the peptide MOA acting through VP1 binding and not induction of a cellular signaling pathway. Although not definitive, it is worth noting that individually D1_min_ residues Trp or Leu to Ala in D1_min_ peptide APQ*W*MLP***L***LLGLY results in loss of antiviral activity and ~1000-fold reduction in potency in vitro. That a mammalian cell-surface receptor would bind the exact same sequence with high affinity and also be susceptible to these same mutations is unlikely, but this MOA cannot be fully excluded based on sequence analysis or sequence changes alone. In summary, the time-of-addition, MOI-dependency, peptide-cell binding, and RT-qPCR studies are convincing and are consistent with an antiviral MOA directly involving VP1 binding.

To the final reviewer’s concern regarding the timing of peptide treatment, we performed an additional cell-based assay comparing the antiviral efficacy of D1_min_ whether the virus or the cells are treated first (“pre-treatment”). We compared our original protocol of pre-treating cells with peptide to pre-treating virus with peptide for the same duration, and found no difference in peptide antiviral EC_50_ (Figure 5—figure supplement 3). This result is consistent with association of the peptide with virus prior to endocytosis.

In summary, we added the following data in response to this revision: Figure 4C, and Figure 5—figure supplements 1-3; the Results section is updated to describe these experiments and their results in the relevant locations. In addition, we discuss the topic of alternative peptide MOA in the second paragraph of the Discussion.

2) Site of Peptide Activity. Experiments show clearly that endocytosis and delivery to the ER is not affected by the peptide, but subsequent translocation of VP1 to the cytosol is disrupted and VP2 is not exposed in the ER. However, it's unclear from these experiments that the peptide is actually present along with the virus in the ER because the antibody to VP2 recognizes the endogenous VP2. Similarly, it's unclear how and when the peptide would engage the virion given how the experiments were conducted. One has to imagine that the peptide is endocytosed along with the virus (or penetrates the membrane separately – similar to the proposal for the Tat-tagged peptide). What's missing is evidence to show peptide and virus engaged within the cell. Finally, can the time-of-addition experiments be explained by a simple model that the authors have not considered, in which the peptide has to bind to the virion, so that adding it after the virus has entered the cell will be too late? (even if this is true, it does not refute the conclusion that the peptide acts at/before disassembly).

This important point relates directly to the cellular MOA. We agree with reviewers that the current experiments do not directly test whether the peptide is associated with the virus after endocytosis. D1_min_ must interact with the virus for delivery into the cell, and the results with the cell-penetrating TAT-fused peptides in our time-of-addition assay are consistent with a peptide antiviral mechanism acting within the cell. Together, these results suggest a mechanism where D1_min_ peptide associates with virions extracellularly and has subsequent antiviral activity within the cell through a mechanism directly involving VP1, or through release of the D1_min_ peptide and subsequent interaction with an intracellular protein. We observe that after endocytosis, peptide D1_min_ inhibits exposure of virion-associated VP2/3. Since virion-associated VP2/3 contains the D1_min_ sequence (within the D1 region of VP2/3), one can infer the target should bind both D1_min_ and VP2/3. Accordingly, direct interaction of D1_min_ and related TAT-fused peptides with pentameric VP1 and subsequent blocking of the interaction of virion-associated VP2/3 with VP1 is a potential intracellular MOA. However, we cannot fully rule out an MOA involving intracellular release of D1_min_ from the virion and subsequent interaction with a cellular target involved in native VP2/3 interactions. Either MOA is revealing and interesting, and further study is required to fully understand the peptide MOA. To clarify the intracellular MOA, we modified the Discussion accordingly.

We indeed attempted to directly assess whether D1_min_ peptide is engaged with the virus within the cell. We performed a colocalization assay with *N*-terminal biotinylated D1_22_ peptide, testing whether colocalization with BKV occurred at various timepoints post-infection. However, while we observed the desired colocalization at various timepoints, we also observed high background due to staining of endogenous cellular biotin, reducing the utility of the assay. We consider these initial results promising but are not including these data in the revised manuscript. We dismissed other methods of assessing peptide-virus interaction, such as affinity purification of biotinylated peptide using streptavidin beads, as they would not demonstrate peptide was bound to virus prior to cell lysis (after which association could occur and give an erroneous positive result).

To address the question regarding the mechanism by which the D1_min_ peptide may enter the cell, the reviewers suggest a model we did not explicitly state, that the peptide must bind the virion in order to enter the cell. It is clear from the time-of-addition study that the peptide must enter the cell contemporaneously with the virus, and that D1_min_ entering the cell is linked to the mechanism of action. We demonstrated that biotinylated peptide is able to bind to infectious virions (Figure 4E), and we favor a mechanism by which the peptide binds to incoming virions to enter cells. Consistent with this model, our peptide-cell binding assay of biotinylated D1_22_ and BK VLPs shows that peptide association with RPTE cells requires interaction with the BK VLPs, and the binding of peptide to VLPs is specific as biotinylated-D1_22_ can be competed away with unlabeled D1_min_ peptide but not mutant D1_min_ L297A peptide. Finally, we’ve updated the text concluding the time-of-addition description to state: “Thus co-entry with the infecting virion, consistent with direct binding to the VP1 pentamer pore, is required for D1_min_ entry into cells and subsequent antiviral activity.”

In summary, we have updated the conclusion of the Results section describing the time-of-addition studies to describe how the peptide likely binds the virus to co-enter cells with BKV during endocytosis, and discuss the lack of direct assay to test BKV and peptide engagement in the cell at the end of the first paragraph of our Discussion section. We additionally describe how the TAT-fused peptides support an intracellular antiviral MOA for D1_min_ in the first and second paragraphs of the Discussion section.

3) Strengthening the Proposed Mechanism of Action. Several simple experiments would significantly strengthen the conclusions.A) VP1 Release/Virion Disassembly. The Helenius group reported a simple gel-based assay for release of VP1 from the capsid (Cell, 2007). It would be informative to determine if the peptide blocked this step. This is important because the only other evidence for a disassembly block is based on immunofluorescence staining in infected cells, which could be misleading.B) Peptide Binding and Function. There is a correlation between peptide binding and inhibitory activity, but there are simple experiments that would greatly strengthen this correlation. For example, the VP1 V234L mutation has little effect on infectivity, and the V234I mutation causes an intermediate phenotype. Peptide binding should be checked in these mutants. If binding is defective, the prediction is that the peptide would not inhibit these mutants. Such a result would definitively establish that binding to the mapped site is required for inhibition. Similarly, do the biotinylated peptides inhibit infection? Does only the N-terminal biotinylated peptide inhibit, which is the one that binds? Finally, the correlation between binding affinity to VP1 and inhibitory activity of mutant peptides was established with purified VP1 pentamers. Affinities should be measured with intact virions, which were used in the infectivity assays. Collectively, these experiments would strengthen the correlation between binding and activity.

We agree with reviewers that more experiments would help provide a more comprehensive description of the peptide antiviral MOA. The gel-based VP1 release / capsid disassembly assay developed by the Helenius group referenced by a reviewer was indeed performed for this study. We performed the assay as described using BKV that was chemically labeled with biotin. We found no effect on capsid disassembly from wild-type D1_min_ treatment; however, the assay lacks a positive control to ensure the quality of the experiment, and as such, we viewed the data as not yet fit for publication. We would add that not all the data provided for the MOA of the peptide is based on immunofluorescent staining. Our fractionation assay is immunoblot-based and shows that the peptide prevents the virion from successfully completing retrotranslocation from the ER lumen into the host-cell cytosol, an important step in polyomavirus entry.

For the proposed additional peptide-binding experiments to bolster the observed relationship between peptide binding and antiviral activity, we performed a limited mutagenesis study involving BKV VP1 residues P232 and V234 and made several mutants (S, L, and I). As described, all three P232 mutations in BKV VP1 resulted in a complete loss of Tag-positive cells (and therefore BKV replication), and mutation V234S also resulted in a complete loss of BKV infectivity. These results show that these pore residues are necessary for replication, in agreement with the Atwood group findings in JCV. With respect to pore mutants V234L and V234I, the substitution of the valine side chain for structurally similar non-polar side chains from leucine and isoleucine had less impact on replication, whereas a switch in polarity from non-polar valine to polar serine had significant impact.

To further explore the link between the infectivity assay and the biochemical assay, we examined one of the V234 mutants, but not both in the AlphaScreen assay. The V234I mutant showed a modest increase in binding response with the D1_22_ peptide compared to wild-type (~2.5-fold, noted in the relevant Results section). This response demonstrates that this mutant is competent for binding VP1 in the biochemical assay, which is consistent with the partial impact on infectivity for the corresponding mutation in BKV. To highlight this datapoint, we’ve updated the relevant figure (Figure 2C) to include VP1 V234I. Again, as discussed in the manuscript, the dataset suggests that while hydrophobic to hydrophilic substitutions that alter the hydrophobicity of the putative peptide binding site (e.g. V234S) have a significant effect on both peptide binding and viral infectivity, substitutions that alter the size of the residue without affecting the pocket hydrophobicity (V234L, V234I) are less impactful. These side chains are flexible, and may be able to rotate into positions that accommodate the respective mutations. As such, it is unsurprising that the V234L substitution does not impact the infectivity of BKV in our spreading infection assay. Speaking to the intermediate phenotype of the V234I substitution in the spreading infection assay, if indeed this mutation perturbs ligand binding to the VP1 pore toward increased affinity relative to the wild-type protein, this could certainly affect viral biology, given that strengthening an interaction can have a deleterious effect on function in addition to weakening an interaction. Additional biochemical studies with the VP1 pore mutants would indeed provide additional insight, but we chose to focus on other experiments related to the D1_min_ peptide.

With respect to reviewer questions regarding the biotinylated peptides used in the current study, we observe both *N*- and *C*-terminal biotinylated D1_22_ (22-mer) peptide elicit antiviral activity against BKV with EC_50_ values that are similar to untagged D1_22_.

The proposal to measure *K*_D_s for the interaction of the peptides with intact virions is an interesting proposal. We know that the D1_22_ peptide with *N*-terminal biotin interacts with BK virions from pull-down experiments, but we did not measure the affinity of this interaction due to the limitations of our methods, which are designed for recombinant proteins (AlphaScreen and Biacore SPR). Measurement of affinity requires the development of a distinct assay with sufficient sensitivity. It is possible that a radioimmunoassay could be developed or perhaps the AlphaScreen assay could be adapted for this purpose, but we are not equipped to run the former, and we are no longer capable of running the latter.